# FE Modelling and Simulation of the Size Effect of RC T-Beams Strengthened in Shear with Externally Bonded FRP Fabrics

**Amirali Abbasi, Zine El Abidine Benzeguir, Omar Chaallal * and Georges El-Saikaly**

Department of Construction Engineering, École de Technologie Supérieure, University of Quebec, Montreal, QC H3C 1K3, Canada; amirali.abbasi.1@ens.etsmtl.ca (A.A.); zine-el-abidine.benzeguir.1@ens.etsmtl.ca (Z.E.A.B.); georges.el-saikaly@etsmtl.ca (G.E.-S.)
* Correspondence: omar.chaallal@etsmtl.ca

**Abstract:** The objective of this study is to conduct a finite-element (FE) numerical study to assess the effect of size on the shear resistance of reinforced concrete (RC) beams strengthened in shear with externally bonded carbon fibre-reinforced polymer (EB-CFRP). Although a few experimental studies have been done, there is still a lack of FE studies that consider the size effect. Experimental tests are time-consuming and costly and cannot capture all the complex and interacting parameters. In recent years, advanced numerical models and constitutive laws have been developed to predict the response of laboratory tests, particularly for issues related to shear resistance of RC beams, namely, the brittle response of concrete in shear and the failure modes of the interface layer between concrete and EB-CFRP (debonding and delamination). Numerical models have progressed in recent years and can now capture the interfacial shear stress along the bond and the strain profile along the fibres and the normalized main diagonal shear cracks. This paper presents the results of a nonlinear FE numerical study on nine RC beams strengthened in shear using EB-CFRP composites that were tested in the laboratory under three series, each containing three sizes of geometrically similar RC beams (small, medium, and large). The results reveal that numerical studies can predict experimental results with good accuracy. They also confirm that the shear strength of concrete and the contribution of CFRP to shear resistance decrease as the size of beams increases.

**Keywords:** size effect; reinforced concrete beams; finite-element method; shear strengthening; externally bonded carbon fibre-reinforced polymer (EB-CFRP) composites

## 1. Introduction

In the last two decades, very few FE studies have been dedicated to RC beams strengthened in shear EB-FRP or any types of strengthening with composite materials [1] made of CFRP [2,3]. However, given the lack of accurate constitutive laws at that time, these early FE studies did not consider the bond between concrete and FRP, nor did they simulate the interaction between concrete and steel reinforcement [4–11]. Recently, some FE studies have concentrated on shear strengthening using embedded-through-section (ETS) and near-surface-mounted (NSM) techniques [12–14]. With recent advances in the development of high-performance FE programs and constitutive laws, numerical studies can better simulate and accurately predict the outcome of experimental tests in terms of load-deflection response, behavior of the interface between concrete and EB-FRP, and the strain distribution along fibres [13,15–35]. Nevertheless, among these studies, very few have considered either the size effect of EB-FRP-strengthened RC beams [33,34] or the crack band model along with the concrete smeared crack model. This was the main impetus to carry out this study to assess the size effect by means of a numerical approach, by implementing both crack models in modelling the concrete and by considering the interface behavior between EB-FRP and the concrete substrate.

Given their complex behavior under loading, as well as their brittle rupture without warning, shear failure in RC beams has long been a major concern in structural engineering.

Therefore, practicing engineers often privilege the sequence by which flexural failure occurs before shear failure. Lack of shear strength in RC beams can be due to various interacting factors. Neglecting the size effect in codes and guidelines and thereby overestimating the ultimate shear capacity in the design process is an example of the effect of such factors. In recent years, the trends towards using FRP composites for rehabilitation and strengthening of RC beams have become intensified due to the high strength-to-weight ratio and tensile strength of FRP composites, which can compensate for the shear-strength deficiency of existing RC beams. It has been established that when the beam size is increased, the shear strength decreases due to the so-called size effect [36–38]. Many parameters affect the size effect, either mitigating or amplifying it by controlling the width of the diagonal shear crack—for example, the rigidity of FRP sheets [39]. Even though comprehensive studies have been performed on the effect of size in RC beams, research studies related to the size effect on RC beams strengthened with EB-FRP are limited. FE analysis can be implemented instead of experimental testing to obtain an inside view of the shear-stress profile variation along the interface layer and the distribution of stress on the fibres during loading. Most analytical models proposed by codes and guidelines are based on experimental results and can be prone to errors (human error, defects in laboratory machines, restricted tools...). Therefore, the results obtained from these models are not as accurate as those from FE methods for capturing the shear resistance contribution of concrete and EB-FRP through appropriate evaluation of strain distributions on the fibres.

The FE method is a cost-effective and precise tool for replacing experimental tests as long as the models are simulated based on reliable and logical assumptions. A few FE studies have been performed on the size effect of RC beams strengthened in shear with EB-CFRP, but either their assumptions were very simplistic, such as perfect bonding between concrete and EB-CFRP, which does not reflect the response of such a beam (location of the shear crack), or they fail to mention the assumptions used in their simulations. As explained in the following sections, the developed 2D-FE model was preferred to 3D models because it is less time-consuming and simulates the propagation of the shear crack in concrete with higher precision. Note that the shear crack is a major parameter in predicting the size effect.

As illustrated in Figure 1, the shear contributions of EB-FRP predicted by ACI 440.2R 2017 for over 50 beams with different depths varying from 80 mm to 682 mm strengthened in shear with continuous U-wrap and strips were compared with their corresponding experimental tests (see Appendix A Table A1 for details). The beams were classified into three categories depending on their depth (Figure 1). As the depth of the specimens and their corresponding EB-FRP bond lengths increased, the ACI 440.2R (2017) guidelines clearly overestimated the shear contribution of EB-FRP, which may indicate the existence of an additional size effect due to the contribution of EB-FRP to shear resistance. In fact, the models of most guidelines overestimate the contribution of EB-CFRP to shear resistance in large specimens.

In the current study, nine RC-T beams without steel stirrups [39] were selected for simulation. The beams were grouped into three series (small, medium, large). In each series, one beam was considered a control (not strengthened with EB-FRP), and the others were strengthened with one and two layers of EB-FRP. The results from the simulated models were validated with experimental tests.

The objectives of the present study were to evaluate the size effect and the shear contributions of concrete and EB-FRP, as well as the effect of an increase in EB-FRP rigidity, on the three series of specimens (different sizes) through numerical investigation. Capturing the response of the interface layer between concrete and CFRP sheets, as well as the distribution of strain along the main fibre of CFRP fabrics during loading, is of paramount importance when using FEA, given their impact on the size effect. Therefore, the impact of the response of the interface layer, the strain distribution along the fibre, and the fibres intercepted by the main diagonal shear crack on the size effect will be studied carefully, along with the failure modes, the load-deflection response, and the pattern of shear cracks. The novelty of this study is to conduct FE research on the size effect and to show the

development of the shear stress and strain in the interface layers and fibres during the loading process. Furthermore, by extracting the strain distribution curve on the fibres that intercepted the main shear crack, it would be possible to measure the distribution factor leading to the effective strain experienced, which is far lower than the effective strain introduced in codes and guidelines.

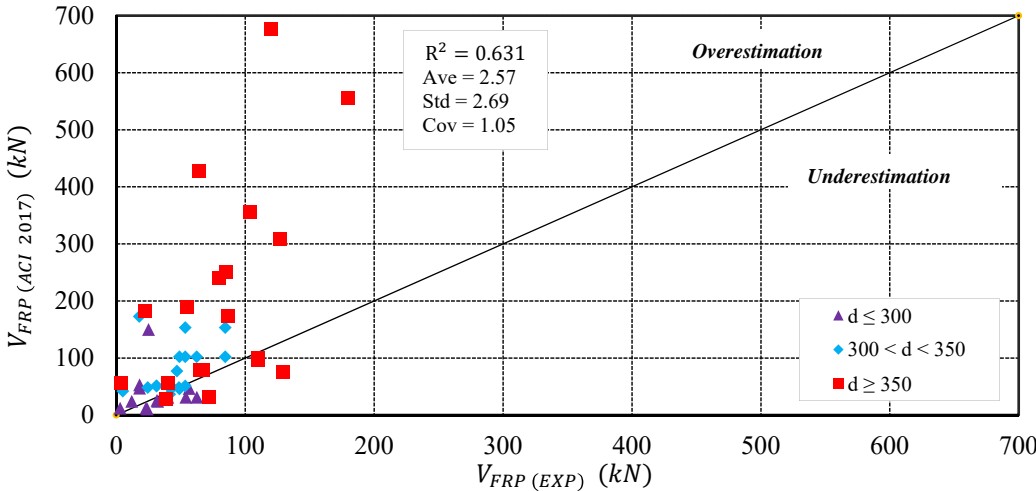

**Figure 1.** Comparison of the predicted ACI 440.2R 2017 code and experimental results.

## 2. Finite-Element Modelling

### 2.1. Suggested FE Modelling

The assumptions implemented for a simulation related to the types of crack models for concrete, steel bar, CFRP sheets, and the interface layer between concrete and CFRP are described in the following sections. Because the beam was not under stress in the normal direction to the plane of the beam, the plane stress model was used for concrete. Steel bars and CFRP sheets were modelled by 2D truss elements that contributed to transferring the stress in the direction of the truss element. Dynamic implicit analysis was implemented to overcome the convergence problem. Indeed, because the divergence occurred due to the brittle behavior of concrete and the nonlinearity of the interface layer between concrete and CFRP (delamination and debonding), general static solvers (static, general and static, Riks) cannot capture the nonlinearity of materials during imposed targeted displacement. Details of the implicit dynamic analysis implementation are described in [40].

### 2.2. Constitutive Models of Materials

#### 2.2.1. Concrete Cracking Models

Various types of concrete cracking models can be used with FEA. The discrete crack model, the rotating smeared crack model, and the fixed smeared crack model are some examples. Considering the discrete crack model, a crack is introduced into the model geometry, where crack propagation occurs along the border of the element in FEA, proving its mesh objectivity. Furthermore, the location of the crack in the model must be defined in advance, which shows the dependency of this technique on how the precise initiation of the crack is predicted. Unlike the discrete crack model, there is no need to predefine the cracking initiation location in the smeared crack model because probable cracking zones and directions are recognized through the smeared crack technique. Elements lose their stiffness as the crack propagates in the smeared crack approach, whereas the stress-strain relation in concrete considers cracks a continuum and predicts the deletion of elements when a crack path is detected. The smeared crack approach can be classified into two categories: the rotating smeared crack approach and the fixed smeared crack approach. The differences between them are their theories for crack direction and their shear retention factor. The deficiency of the smeared crack model is that when element size decreases,

it leads to zero energy dissipation in the softening part of the stress–strain curve in the tensile concrete material, resulting in strain localization [35]. To address strain localization, some limiters have been proposed, among which the crack band model implemented in the concrete damage plasticity framework has been proved to address mesh objectivity challenges resulting in convergence problems [41]. The function of the crack band model is to convert the width of the crack band to the cracking strain caused by the crack and softening behavior of the concrete in the tension, as shown in Figure 2.

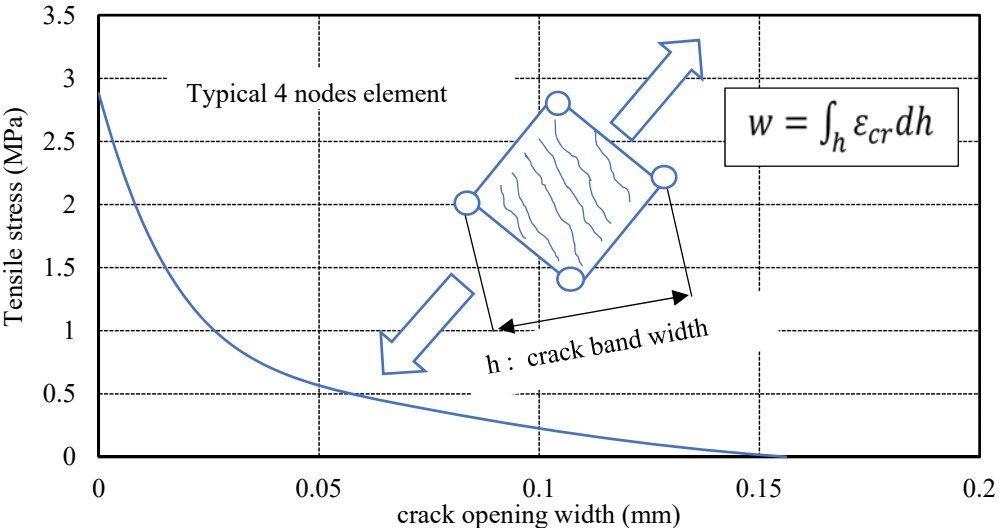

**Figure 2.** Relation between cracking width and tensile stress adopted from [42].

### 2.2.2. Concrete Response in Compression and Tension

Because the RC T-beams in the present study behave in their plane, a four-node plane stress element (CPS4) was implemented to simulate concrete. Various models have been proposed to represent the uniaxial behavior of concrete in compression, among which the model introduced by [43] (see Equation (1)) features a reasonable prediction of the ascending and softening parts of the concrete material curve, as shown in Figure 3.

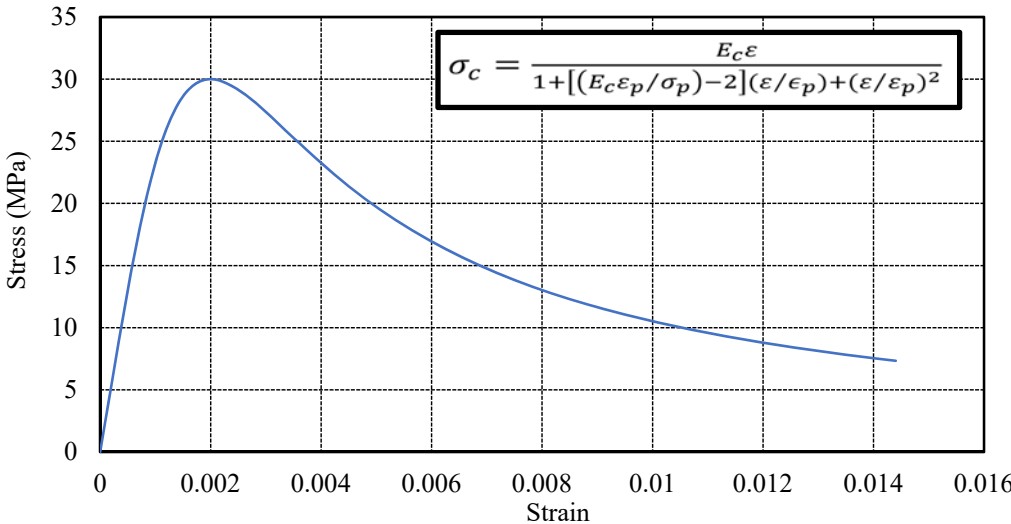

**Figure 3.** Stress–strain model for uniaxial compression in concrete introduced by [43].

$$\sigma_c = \frac{\alpha \varepsilon}{1 + \left[ \left( \alpha \varepsilon_p / \sigma_p \right) - 2 \right] (\varepsilon / \epsilon_p) + \left( \varepsilon / \varepsilon_p \right)^2} \tag{1}$$

where $\sigma_p$ and $\varepsilon_p$ are the maximum concrete compressive stress and strain obtained from experimental tests, equal to $f_c' = 30$ MPa and 0.002, respectively; $E_c$ is the concrete modulus of elasticity $E_c = 4730\sqrt{f_c'}$ (MPa) according to [44]; and $\sigma$ and $\varepsilon$ are the applied compressive stress and corresponding strain during loading of the cylindrical specimen, respectively.

To define tensile concrete behavior in the descending and softening parts, the model introduced by [42] on the basis of numerous stress-crack displacement tests was implemented in this study as follows:

$$\frac{\sigma}{f_t} = \left[1 + \left(c_1 \frac{w}{w_{cr}}\right)^3\right] e^{\left(-c_2 \frac{w}{w_{cr}}\right)} - \frac{w}{w_{cr}}\left(1 + c_1^{\,3}\right)e^{-c_2} \tag{2}$$

$$w_{cr} = 5.14 \frac{G_f}{f_t} \tag{3}$$

$$f_t = 1.4\left(\frac{f_c' - 8}{10}\right)^{\frac{2}{3}} \tag{4}$$

$$G_f = \left(0.0469 d_a^{\,2} - 0.5 d_a + 26\right)\left(\frac{f_c'}{10}\right)^{0.7} \tag{5}$$

where $w$ is the crack width during loading; $w_{cr}$ is the crack width at the moment when no stress can be transferred between the two sides of the crack; $f_t$ is the maximum concrete tensile stress; $\sigma$ is the tensile stress in the specimen during the stress-crack displacement test; $G_f$ is the fracture energy, which in addition to Equation (5) can be obtained from the area of the stress-cracking displacement graph (Figure 2); $c_1 = 3$ and $c_2 = 6.93$ are constant parameters proposed by [42]); and $d_a$ is the largest aggregate dimension.

### 2.2.3. Definition of Compressive and Tensile Damage to Concrete Damage Plasticity (CDP)

To define concrete damage in both compression and tension, represented by the softening part of the stress–strain curves, the proposed model introduced by [45] was considered as follows:

$$d_{t,c} = \begin{cases} \frac{(1-k)\varepsilon^P}{(1-k)\varepsilon^P + \sigma/E_0} & if \ \dot{\bar{\varepsilon}}^P \geq 0 \\ \frac{\varepsilon^P - (\varepsilon - \bar{\varepsilon}_{cr}^e)}{\varepsilon^P - (\varepsilon - \bar{\varepsilon}_{cr}^e) + \sigma/E_0} & if \ \dot{\bar{\varepsilon}}^P < 0 \end{cases} \tag{6}$$

where $d_{t,c}$ is the damage parameter in both tension and compression, $\dot{\bar{\varepsilon}}^P$ is the plastic strain rate, $k$ is the rate of inelastic strain when stiffness degrades ($\bar{\varepsilon}^P$) to inelastic strain when stiffness is constant ($\varepsilon^P$), and $\bar{\varepsilon}_{cr}^e$ is the cracking strain when the plastic strain rate is zero. The smeared crack model is implemented in the concrete damage plasticity (CDP) framework. Therefore, the stress–strain behavior of concrete in tension is transformed to stress-cracking displacement through the crack band model $\varepsilon_t^p = w_t/h$ [45]. Furthermore, the graphs in Figure 4a,b obtained from Equation (6) are applied for both tensile and compressive damage in concrete versus cracking displacement and plastic strain, respectively (for 10 mm element size). It has been proven that concrete damage plasticity (CDP) is able to show the response of concrete with high accuracy even for different types of concrete beams such as precast segmental concrete beams [46,47].

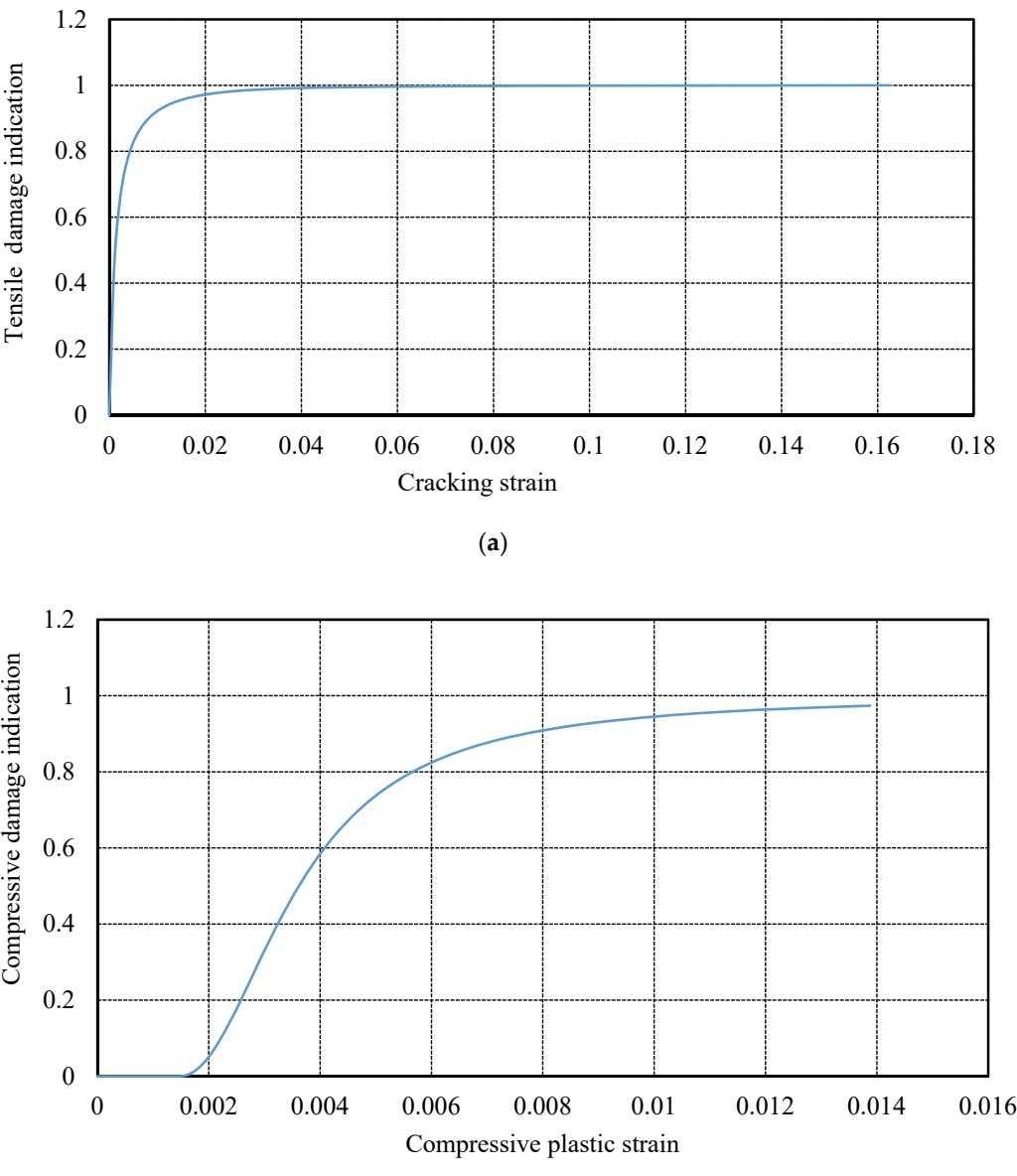

**Figure 4.** Tensile concrete damage model (for 10 mm element size): (**a**) compressive concrete damage models; (**b**) proposed by [45].

### 2.3. Bond-Slip Model for Concrete–Steel Reinforcement and Concrete–CFRP

To predict the ultimate shear capacity of an RC beam shear-strengthened with EB-FRP, the interaction between concrete and FRP composites should be defined precisely; otherwise, the software cannot identify the potential failure modes between concrete and CFRP, such as debonding and delamination. Early simulations assumed a perfect bond between the components of such beams, resulting in overestimation of the load-carrying capacity of the specimens. In addition, the perfect bond model between concrete and EB-CFRP has an effect on the distribution, direction, and position of shear cracks, leading to incorrect debonding and delamination. Because no slips were observed between concrete and longitudinal bars, a perfect bond model was assumed between the concrete and the longitudinal steel reinforcement. As for the bond between the concrete substrate and EB-CFRP, a two-dimensional, four-node cohesive element (COH2D4) that could capture both debonding and delamination failures in the model was implemented in ABAQUS. To define the properties of the cohesive elements, a simplified bond–slip law introduced

by [48] was implemented in this study (Figure 5). The ascending and softening parts of the bond–slip curves were defined as follows:

$$\tau = \tau_{max}\sqrt{\frac{s}{s_0}} \; if \; s \leq s_0 \tag{7}$$

$$\tau = \tau_{max}e^{-\alpha\left(\frac{s}{s_0}-1\right)} \; if \; s > s_0 \tag{8}$$

where $s_0 = 0.0195\beta_w f_t$, $G_f = 0.308\beta_w^2\sqrt{f_t}$, $\alpha = \dfrac{1}{\frac{G_f}{\tau_{max}s_0}-\frac{2}{3}}$, $\beta_w = \sqrt{\dfrac{2-\left(w_f/(s_f sin\beta)\right)}{1+\left(w_f/(s_f sin\beta)\right)}}$, and $\beta =$ fibre orientation. In the direction normal to the cohesive layer, which is representative of interface delamination, the following model was implemented for the cohesive layer to estimate the initial stiffness:

$$K_{nn} = \frac{1}{\frac{t_{concrete}}{E_{concrete}} + \frac{t_{epoxy}}{E_{epoxy}}} \tag{9}$$

where $t_{concrete}$ is the substrate thickness of concrete, $t_{epoxy}$ is the thickness of epoxy, and $E_{concrete}$ and $E_{epoxy}$ are the concrete and epoxy moduli of elasticity, respectively. The maximum tensile strength normal to the cohesive layer was also assumed equal to the maximum strength of concrete in tension.

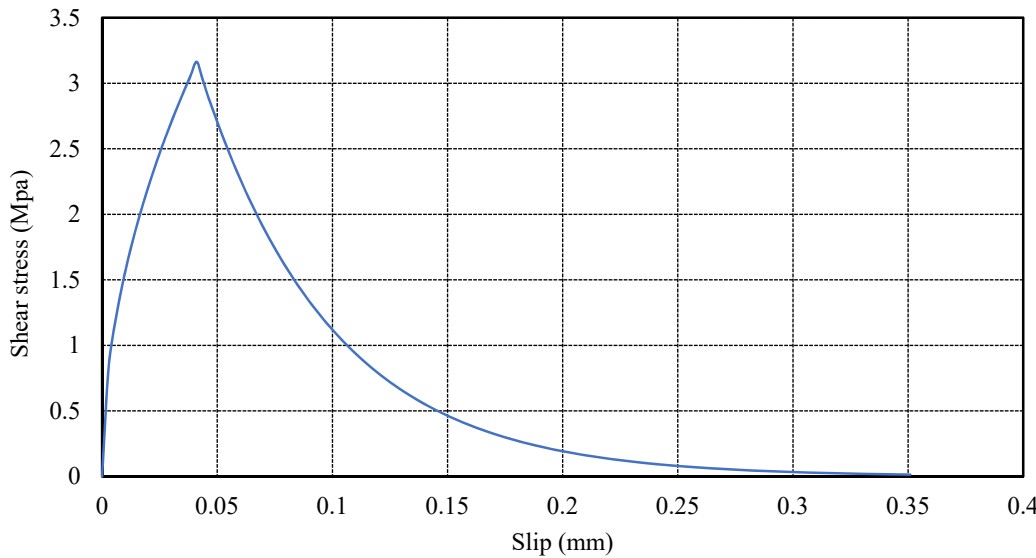

**Figure 5.** Bond–slip model between concrete and CFRP proposed by [48].

### 2.4. Modelling Internal Steel Reinforcement and EB-CFRP

To model the internal steel reinforcement and the external CFRP fabric, two-node 2D truss elements (T2D2) were implemented in the current study. Details of the simulation are illustrated in Figure 6. The elastic–plastic material was assigned to the steel reinforcement where bilinear response of the stress–strain behavior of steel bars in tension was assumed instead of nonlinear behavior after reaching the elastic limit to reduce calculation time (Figure 7a). As for EB-CFRP, the material was considered elastic until rupture in such a way that CFRP fibres could contribute to shear resistance through their tensile strength (Figure 7b) while their compression strength was zero. Based on [40], it was assumed that when FRP wrap is modelled by truss elements, the space between truss elements should be approximately $S_f = \frac{h_{f,e}}{20}$ to achieve reasonable agreement with continuous FRP fabrics. Therefore, the space between the truss elements modelling CFRP fibres was set to 10 mm, 10 mm, and 5 mm for large, medium, and small beams, respectively.

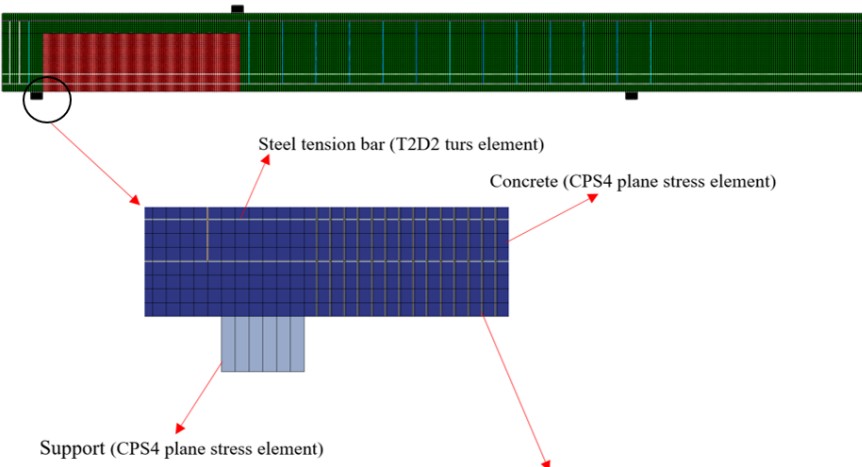

**Figure 6.** 2D simulation of the strengthened RC T-beams and their defined elements in ABAQUS.

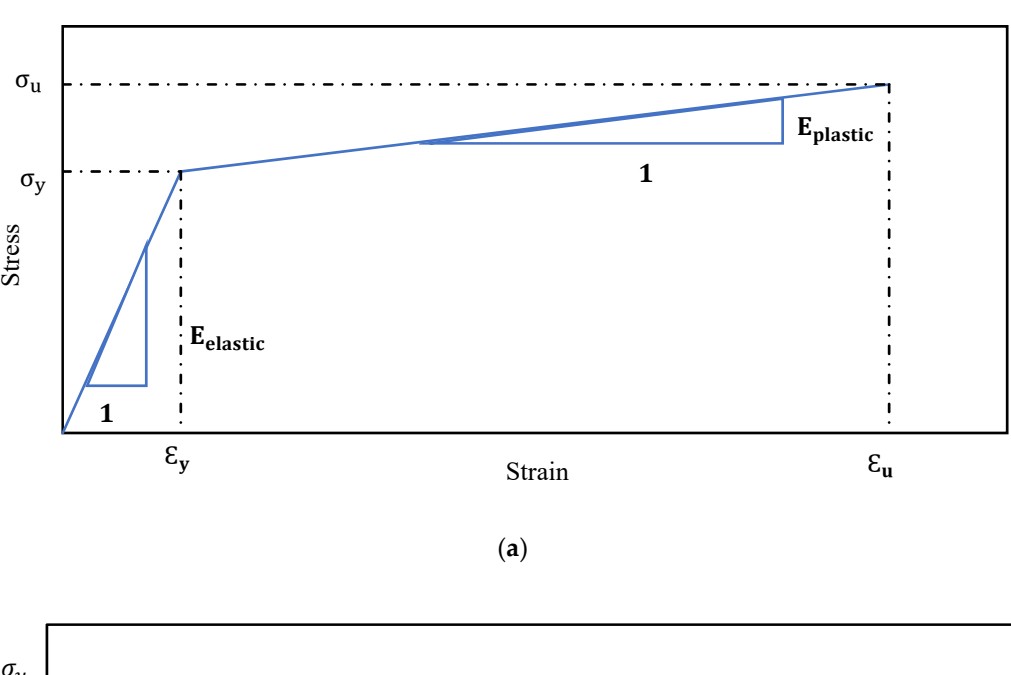

(**a**)

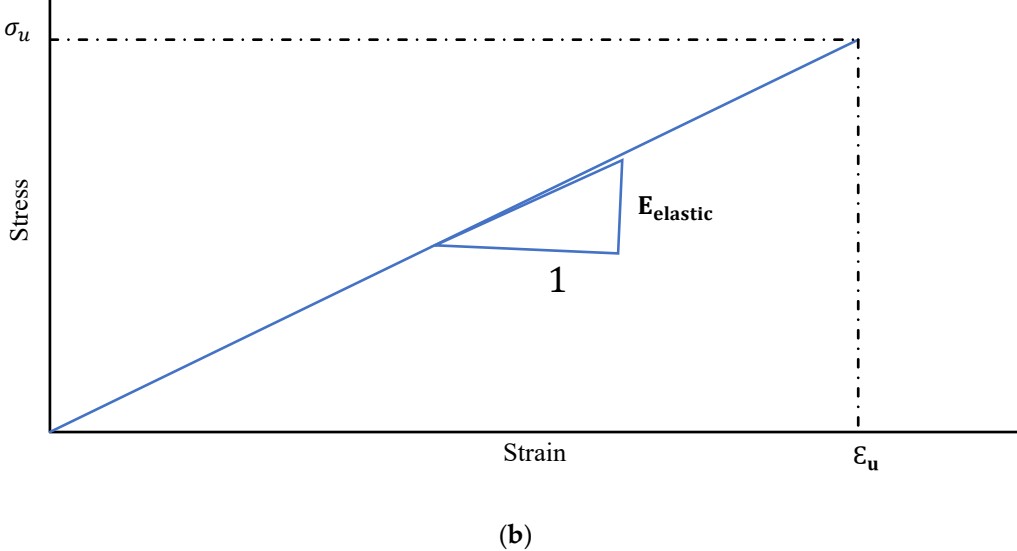

(**b**)

**Figure 7.** Stress–strain relation for (**a**) steel reinforcement and (**b**) CFRP fabrics.

## 3. Experimental Investigation

Nine RC T-beams without steel stirrups were selected from the experimental tests (control and strengthened using EB-CFRP) conducted by [39] to investigate the size effect by means of FEA. In addition, this study assessed the impact of increasing the rigidity of EB-CFRP on its contribution to the shear resistance of RC beams. The results are presented in terms of (1) load-deflection responses, (2) strain profiles along the normalized diagonal shear cracks, (3) strain profiles along the fibre direction, and (4) variation of interfacial shear stress profiles along the cohesive layer. Details of the geometry, steel reinforcement position, and configuration of EB-CFRP are illustrated in Figure 8. These beams were grouped into three series of RC T-beams that were geometrically similar, but of different sizes: large, medium, and small, abbreviated as L, M, and S, respectively. One beam in each series was not strengthened and served as a control beam. The specimens were subjected to a three-point loading scheme. The geometry and properties of the nine selected specimens are presented in Table 1.

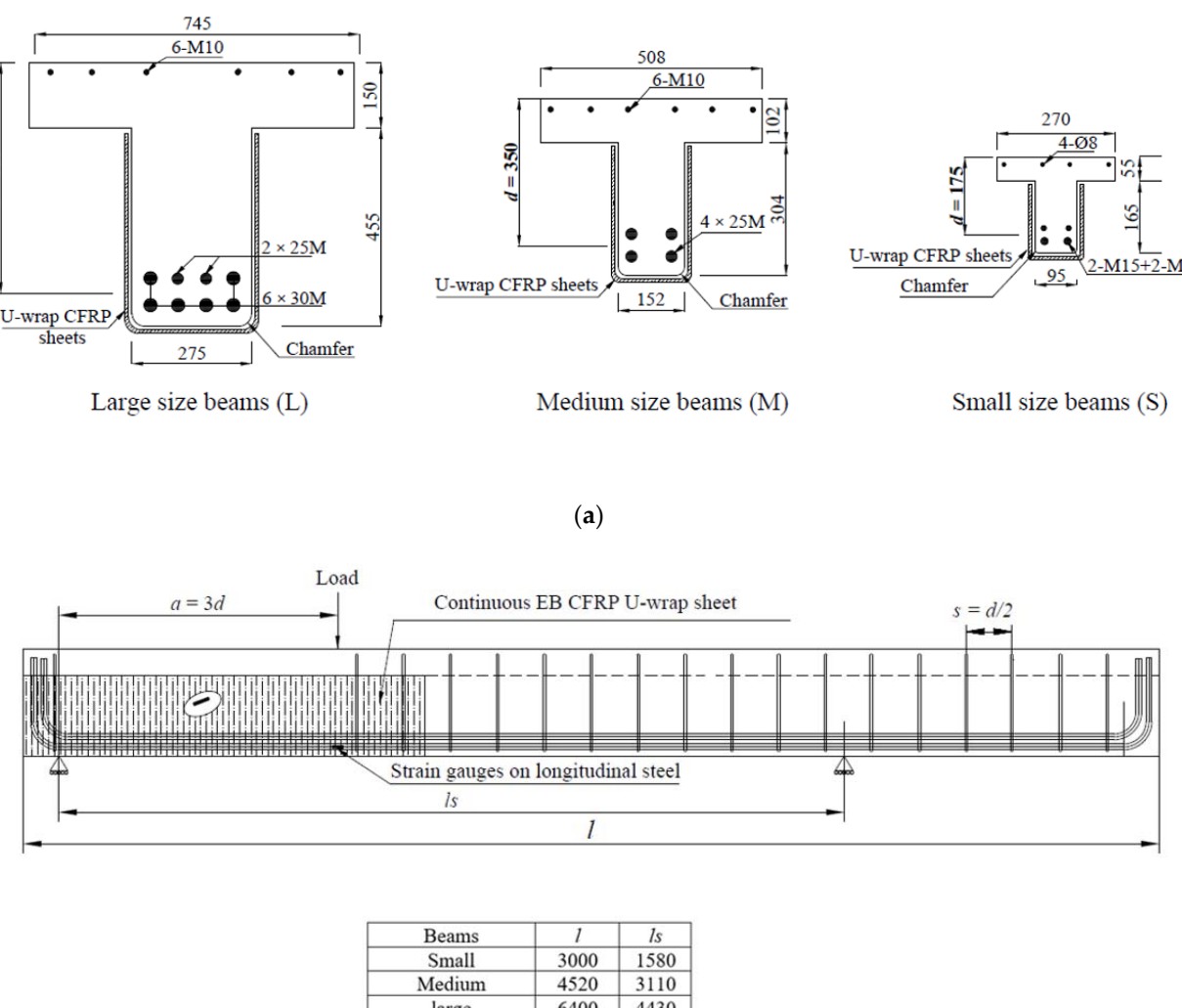

(**a**)

| Beams | *l* | *ls* |
|---|---|---|
| Small | 3000 | 1580 |
| Medium | 4520 | 3110 |
| large | 6400 | 4430 |

*l : total length of beams (mm); ls : distance between supports (mm)*

(**b**)

**Figure 8.** Details of beams: (**a**) cross-sections of large, medium, and small specimens (mm) and (**b**) elevation of beam and position of three-point loading ([39]).

**Table 1.** Geometry and property of material in studied beams.

| | | Series | | | | | | | | |
|---|---|---|---|---|---|---|---|---|---|---|
| | | S0-Con | | | S0–1L | | | S0–2L | | |
| | Specimen | S.S0.Con | M.S0.Con | L.S0.Con | S.S0.1L | M.S0.1L | L.S0.1L | S.S0.2L | M.S0.2L | L.S0.2L |
| Concrete | $f'_c$ (MPa) | 30 | 30 | 30 | 30 | 30 | 30 | 30 | 30 | 30 |
| | a/d | 3 | 3 | 3 | 3 | 3 | 3 | 3 | 3 | 3 |
| | Beam length, mm | 3000 | 4520 | 6400 | 3000 | 4520 | 6400 | 3000 | 4520 | 6400 |
| | Flange height, $h_f$, mm | 55 | 102 | 150 | 55 | 102 | 150 | 55 | 102 | 150 |
| | Flange width, $b_f$, mm | 270 | 508 | 745 | 270 | 508 | 745 | 270 | 508 | 745 |
| | Web height, $h_w$, mm | 165 | 304 | 455 | 165 | 304 | 455 | 165 | 304 | 455 |
| | Web width, $b_w$, mm | 95 | 152 | 275 | 95 | 152 | 275 | 95 | 152 | 275 |
| | Shear span | 525 | 1050 | 1575 | 525 | 1050 | 1575 | 525 | 1050 | 1575 |
| Steel Bars | Tensile bars | 2 × M15 + 2 × M10 | 4 × M25 | 6 × M30 + 2 × M25 | 2 × M15 + 2 × M10 | 4 × M25 | 6 × M30 + 2 × M25 | 2 × M15 + 2 × M10 | 4 × M25 | 6 × M30 + 2 × M25 |
| | Tensile yielding stress, MPa | 420–440 | 470 | 420–470 | 420–440 | 470 | 420–470 | 420–440 | 470 | 420–470 |
| | Modulus of elasticity $E_s$, GPa (T) | 175–200 | 200 | 210–200 | 175–200 | 200 | 210–200 | 175–200 | 200 | 210–200 |
| | Compressive bars | 4 × ϕ8 | 6 × M10 | 6 × M10 | 4 × ϕ8 | 6 × M10 | 6 × M10 | 4 × ϕ8 | 6 × M10 | 6 × M10 |
| | Compressive yielding stress, MPa | 650 | 440 | 440 | 650 | 440 | 440 | 650 | 440 | 440 |
| | Modulus of elasticity $E_s$, GPa (C) | 215 | 200 | 200 | 215 | 200 | 200 | 215 | 200 | 200 |
| CFRP Fabrics | Configuration | - | - | - | Ct-U | Ct-U | Ct-U | Ct-U | Ct-U | Ct-U |
| | Thickness of fabrics, $t_{CFRP}$, mm | - | - | - | 0.066 | 0.107 | 0.167 | 0.132 | 0.214 | 0.334 |
| | Modulus of elasticity $E_f$, GPa | - | - | - | 231 | 231 | 234 | 231 | 231 | 234 |
| | Tensile strength, MPa | - | - | - | 3650 | 3650 | 3793 | 3650 | 3650 | 3793 |
| | Number of layers | - | - | - | 1 | 1 | 1 | 2 | 2 | 2 |

## 4. Validation with Experimental Tests

As mentioned earlier, the simulated model has been validated by the experimental tests carried out by [39]. The element size for discretization of the small beams (one- and two-layer strengthening) and the control beam was 5 mm, 10 mm, and 10 mm for small, medium, and large beams, respectively. These sizes have shown good agreement between numerical and experimental results.

### 4.1. Failure Modes, Crack Patterns, and Ultimate Load-Carrying Capacity

Negative strain shows as compression in concrete, which mainly occurs around the supports and the load plate. Figure 9 shows the numerical results, which illustrate the main shear crack distributions in all strengthened beams by means of the principal logarithmic plastic strain in the plane of the beams. As shown in Figure 9, regardless of size, the patterns of shear cracks for small, medium, and large beams strengthened with two CFRP layers were similar (Figure 9a,c–e), starting in the mid-depth of the web and extending to the support and the web/flange intersection to propagate horizontally towards the load plate. The results for the medium and large strengthened specimens with one CFRP layer (Figure 9b–d) followed the same trend.

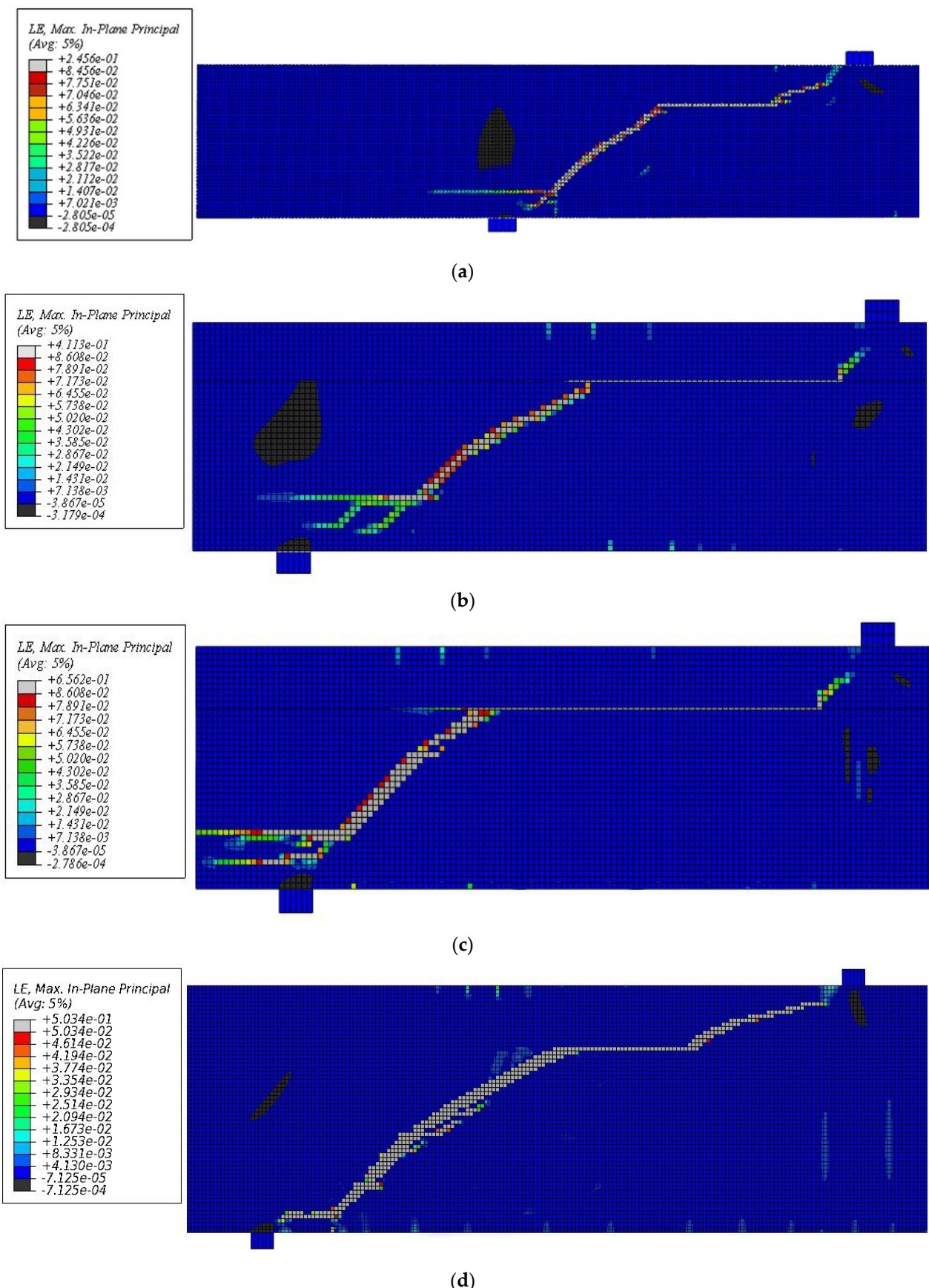

**Figure 9.** *Cont.*

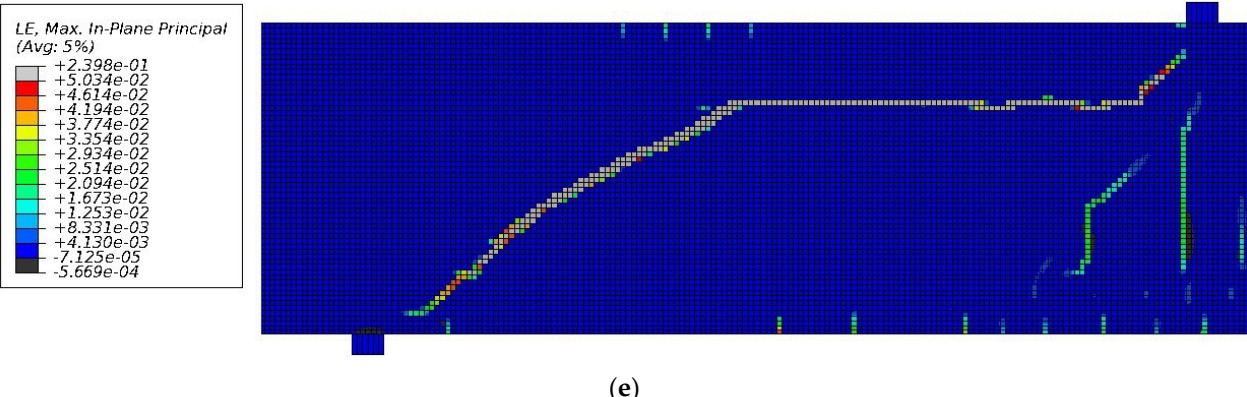

(**e**)

**Figure 9.** Crack pattern obtained from simulation in ABAQUS for specimen at complete failure: (**a**) specimen S.S0.2L; (**b**) specimen M.S0.1L; (**c**) specimen M.S0.2L; (**d**) specimen L.S0.1L; (**e**) specimen L.S0.2L.

Experimental tests of the control beams (small, medium, large) showed similar crack angle patterns with one single diagonal shear crack, appearing as a crack band at mid-height of the web and propagating toward the web soffit (support) and the flange (load application point) of the beam. As shown in Figure 10, the maximum crack angle occurred at mid-depth of the beams and then decreased as the crack extended towards the support and the load application point. The crack patterns predicted through numerical analysis were in good agreement with experimental results (Figure 10a–d). The failure thresholds of the control specimens with increasing load are presented in Table 2.

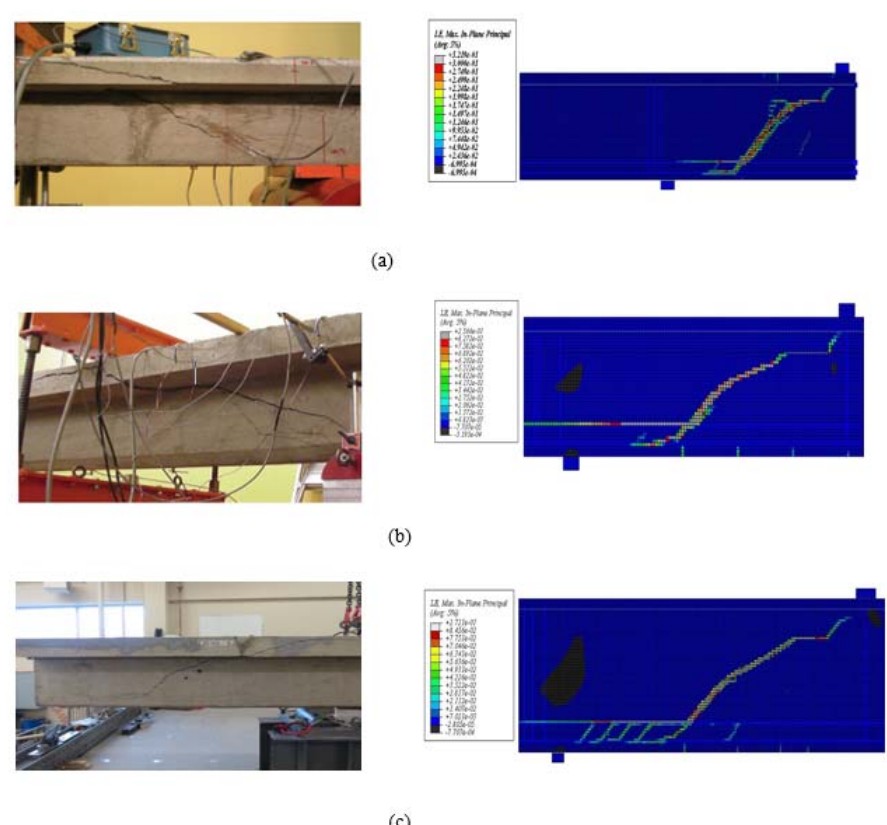

**Figure 10.** Crack pattern obtained from FEA and experimental testing for control beams at ultimate states: (**a**) S.0L.Con; (**b**) M.0L.Con; (**c**) L.0L.Con.

**Table 2.** Failing procedure of control specimens as increasing load for experimental and numerical tests.

| | Imposed Load | | | | |
|---|---|---|---|---|---|
| | **Experimental Test** | | **Numerical Prediction** | | |
| **Specimen** | **Failure Load (kN)** | **Shear Cracks (kN)** | **Failure Load (kN)** | **Shear Cracks (kN)** | $P_{num.}/P_{exp.}$ |
| S.S0.Con | 58 | 19 (31%) | 62 | 25 (40%) | 1.06 |
| M.S0.Con | 130 | 45 (35%) | 128 | 60 (46%) | 0.98 |
| L.S0.Con | 283 | 73 (25%) | 293 | 89 (30%) | 1.03 |

Note that the flexural and shear cracks in the small and medium specimens occurred at approximately the same ratio of the ultimate loads. However, this ratio decreased considerably for large beams, indicating the possible existence of a size effect in large specimens that reduced their shear strength capacities as depth increased. Numerical results showed that the ultimate load of the medium beam was 106% higher than that of the small beam, and that the ultimate load of the large beam was 372% and 128% higher than those of the small and medium ones, respectively. The failure loads occurred at 62, 128, and 293 kN, whereas shear cracks formed at 25 kN, 60 kN, and 89 kN applied loads for small, medium, and large beams, respectively. This was in good agreement with the experimental results (see Table 2). Single diagonal shear cracks formed in control beams (small, medium, and large), giving rise to shear failure in all specimens. This confirms the results obtained by [49]. Moreover, based on the experimental results, the shear crack angles in small, large, and medium beams were 42°, 37°, and 24° respectively, which are comparable with the numerical results (Figure 10a–d).

As shown in Figure 10, crack patterns in all control specimens were similar regardless of specimen size. However, the large beams featured more distributed minor cracks, probably due to wider cracking and the resulting loss of aggregate interlock (Figure 10c). Delamination of the interface layer occurred in all strengthened specimens when the stresses normal to the interface layer exceeded their maximum resistance (2.3 MPa). The delamination started from the top edge of the CFRP wrap located at the web/flange intersection and then extended horizontally and propagated vertically towards the top parts of the main diagonal shear crack. The stress normal to the interface layer at the web/flange intersection exceeded 2.3 MPa, which is the maximum strength in the normal direction of the interface layer.

As beam depths decreased from 525 mm to 350 mm and then to 175 mm, the behavior of the specimens changed from brittle to ductile, as indicated by the load-deflection response with a plateau (Figure 11a). Numerical results commonly overestimate the load-carrying capacity of a beam by approximately 6% because the bond between longitudinal bars and concrete is assumed perfect and an implicit dynamic is implemented to solve the model, thus amplifying deflection and load in the dynamic analysis [35,40]. However, as long as the parameters in the dynamic analysis are defined appropriately (time increment, loading time, and loading scheme), it can be an appropriate replacement for static analysis [35,40].

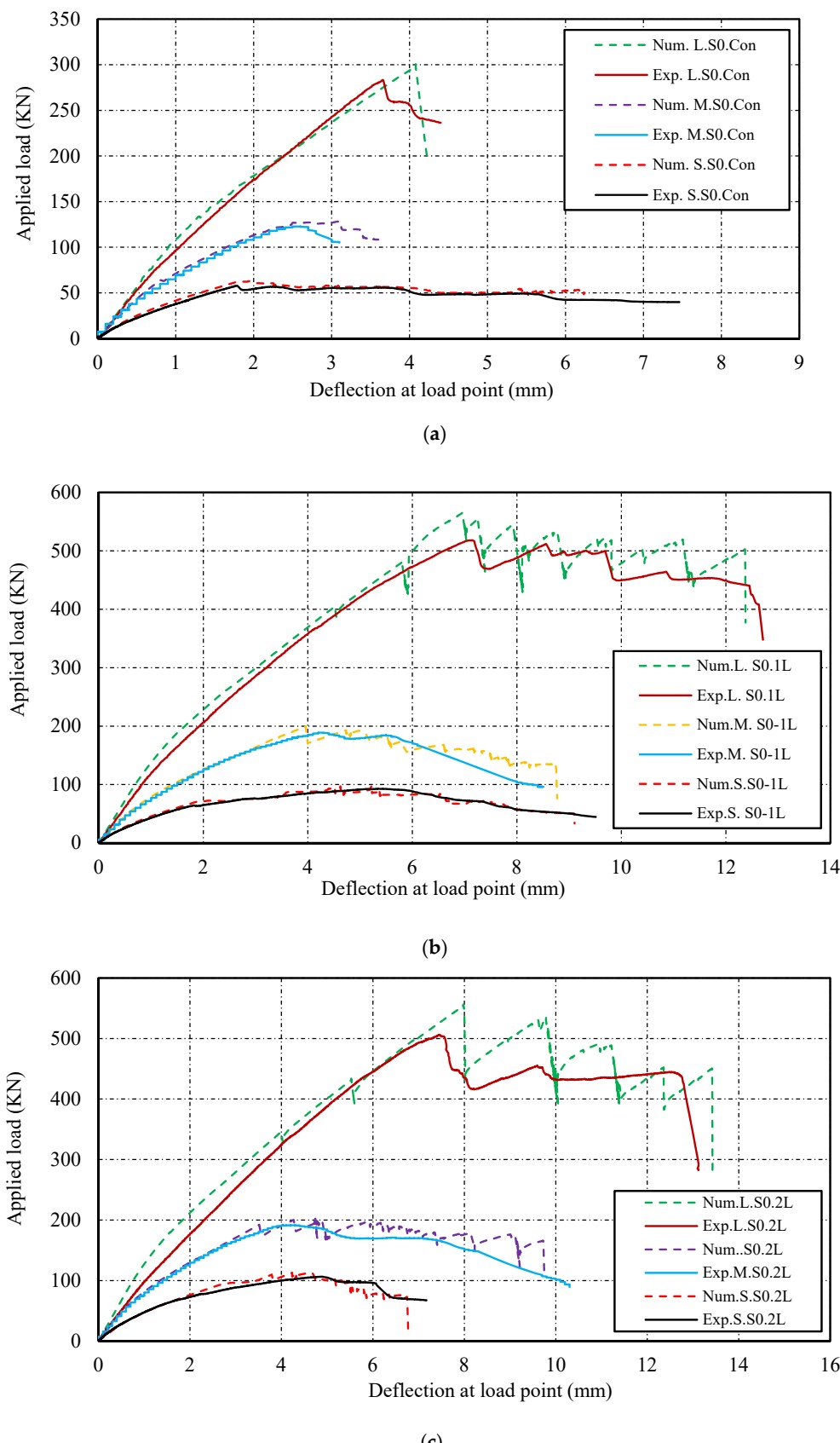

**Figure 11.** Numerical vs. experimental load-deflection response: (**a**) control beams; (**b**) beams strengthened with one CFRP layer; (**c**) beams strengthened with two CFRP layers.

The results of the numerical and experimental tests are summarized in Tables 3 and 4 in terms of ultimate load ($P_{max}$), deflection at $P_{max}$, shear contribution of CFRP ($V_{CFRP}$), shear gain due to CFRP ($G_{CFRP}$), maximum shear force ($V_T$), maximum shear stress in the section $v = V_T/(b_w \times d)$, loss of shear stress in medium and large specimens with respect to the small beam $v$ (%), and ultimate strain contributed by CFRP fabrics in each specimen.

**Table 3.** Comparison between experimental and numerical results in terms of load deflection and ultimate shear strength contributed by concrete and CFRP fabrics.

| Specimens | $P_{max}$ (kN) | | $\Delta_{Pmax}$ (mm) | | $V_T$ (kN) | | $V_{CFRP}$ (kN) | | $P_{num}/P_{max}$ | Failure Mode | |
|---|---|---|---|---|---|---|---|---|---|---|---|
| | Exp. | Num. | Exp. | Num. | Exp. | Num. | Exp. | Num. | | Exp. | Num. |
| S.S0.Con | 58 | 62 | 1.8 | 1.9 | 38 | 41 | - | - | 1.06 | Shear | Shear |
| M.S0.Con | 130 | 128 | 2.6 | 3.1 | 86 | 85 | - | - | 0.98 | Shear | Shear |
| L.S0.Con | 283 | 293 | 3.7 | 4 | 182 | 189 | - | - | 1.03 | Shear | Shear |
| S.S0.1L | 93 | 98 | 5.4 | 5.2 | 62 | 65 | 23 | 24 | 1.05 | Shear | Shear |
| M.S0.1L | 189 | 195 | 4.2 | 3.9 | 125 | 130 | 39 | 45 | 1.03 | Shear | Shear |
| L.S0.1L | 518 | 550 | 7.1 | 7.2 | 334 | 354 | 151 | 165 | 1.06 | Shear | Shear |
| S.S0.2L | 106 | 112 | 4.9 | 4.23 | 71 | 75 | 32 | 34 | 1.05 | Shear | Shear |
| M.S0.2L | 191 | 202 | 4.1 | 4.8 | 127 | 134 | 40 | 49 | 1.05 | Shear | Shear |
| L.S0.2L | 506 | 543 | 7.5 | 8 | 326 | 349 | 144 | 160 | 1.07 | Shear | Shear |

**Table 4.** Comparison between numerical and experimental results in terms of shear gain and loss.

| Specimens | Shear Gain Due to FRP (%) | | Shear Stress in Concrete | | Loss in Shear Stress with Respect to Control Beam (v%) | |
|---|---|---|---|---|---|---|
| | Exp. | Num. | Exp. | Num. | Exp. | Num. |
| S.S0.Con | - | - | 2.31 | 2.46 | - | - |
| M.S0.Con | - | - | 1.62 | 1.59 | 30 | 35 |
| L.S0.Con | - | - | 1.26 | 1.31 | 45 | 47 |
| S.S0.1L | 60 | 58 | 3.71 | 3.91 | - | - |
| M.S0.1L | 45 | 52 | 2.35 | 2.44 | 37 | 38 |
| L.S0.1L | 83 | 87 | 2.31 | 2.45 | 38 | 37 |
| S.S0.2L | 84 | 83 | 4.26 | 4.51 | - | - |
| M.S0.2L | 47 | 58 | 2.38 | 2.51 | 44 | 44 |
| L.S0.2L | 79 | 85 | 2.26 | 2.41 | 47 | 47 |

*4.2. Load-Deflection Responses*

Figure 11 compares the experimental and numerical results in terms of ultimate load-carrying capacities and displacements for the nine specimens. Note that shear strengthening with EB-CFRP fabrics showed higher levels of strength in specimens strengthened with one layer by about 58%, 52%, and 88% for small, medium, and large beams, respectively, with respect to control beams. Furthermore, the deflections corresponding to the maximum load ($\Delta_{pmax}$) of specimens strengthened with one EB-CFRP layer increased by 173%, 25%, and 80% with respect to control beams, which could be attributed to the fact that CFRP fabrics control the deflection of specimens (Figure 11). Nevertheless, by adding two CFRP layers, no considerable additional deflections were observed in the specimens compared to those strengthened with one layer. The results of the load-deflection data obtained from numerical analysis are highly comparable with experimental observations, showing that the simulated model can predict laboratory tests with high accuracy.

## 5. FE Simulations and Results

This section is dedicated to FE simulations and analyses. The results are presented in terms of (a) shear strength for control and strengthened beams, (b) distribution of strain on the fibres along the diagonal shear crack, and (c) strain distributions along the CFRP fabric and interfacial shear stress at the cohesive layer.

### 5.1. Shear Strength and Loss in Control and Strengthened Beams

This section presents the FEA results for shear strength and shear loss due to the size effect. The size effect had an impact on strengthened beams in the way that the shear stress contributed by CFRP fabric decreased in specimens strengthened with one CFRP layer, from 1.45 MPa in S.S0.1L to 1.14 MPa in L.S0.1L. Table 4 compares the numerical and experimental results. The specimens of the third series, which were strengthened by two layers, resulting in higher CFRP rigidity, showed similar results, with shear stress decreasing from 2.05 MPa in S.S0.2L to 1.1 MPa in L.S0.2L. As illustrated in Figure 12, adding a second layer of EB-CFRP increased the shear stress in the small beam before delamination by 30%, that is, from 1.45 MPa to 2.05 MPa. This gain in shear stress decreased in the medium specimen by 7% and in the large specimen by 3%, indicating that the size effect has an impact on the shear stress contributed by both concrete and CFRP. Nevertheless, more investigations are required to clarify the relation between the size effect and the rigidity of CFRP as the specimen dimension increases.

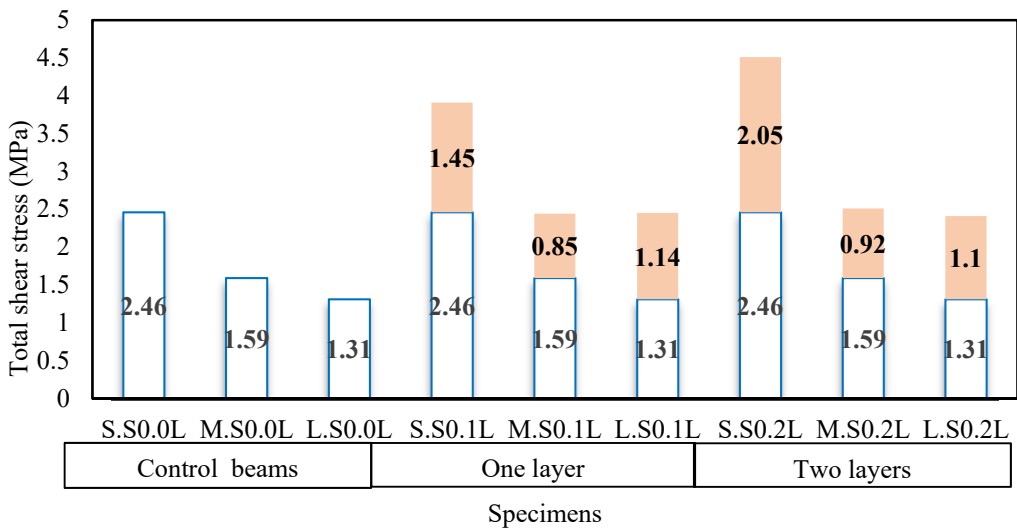

**Figure 12.** Shear stress contributed by concrete and CFRP fabric (FE results).

### 5.2. Distribution of Strain on the Fibres along the Diagonal Shear Crack by FEA

Assessing the strain profile along the main diagonal shear crack resulted in a better understanding of the behavior of fibres and their contribution to shear strengthening as the size of specimens and the rigidity of CFRP fabrics increased. By evaluating the strain distribution along the shear crack path, it is possible to locate the maximum crack width and to understand how debonding and delamination occur on both sides of the crack. The main diagonal shear cracks had an almost linear pattern for specimens strengthened with two layers of CFRP fabric and a semi-parabolic shape for those strengthened with one CFRP layer (Figures 9 and 10, respectively). Because the strain distribution on the fibres along the main diagonal shear crack constitutes the basis on which the distribution factors for the strain are established, the response of the strain in fibres intersected by the normalized shear crack is evaluated in this section. Note that the distribution factor $D_{FRP}$

introduced by [50] is a function of the average strain on the fibres intercepted by the shear crack and the maximum strain experienced by fibres at a specific load, as follows:

$$D_{FRP} = \frac{\sum_{i=1}^{n} \varepsilon_{FRP,i}}{n\varepsilon_{max}}$$

(10)

where $\varepsilon_{FRP,i}$ is the strain in the $i^{th}$ specific fibre intercepted by the shear crack, $n$ is the number of fibres crossed by the diagonal shear crack, and $\varepsilon_{max}$ is the maximum strain experienced by the fibre at ultimate state.

The formula introduced by [51] for the shear contribution of EB-FRP is:

$$V_f = 2f_{f,e} t_f w_f \frac{h_{f,e}(cot\theta + cot\beta)sin\beta}{S_f}$$

(11)

where $f_{f,e}$ is the effective stress in the *FRP* intercepted by the main shear diagonal shear crack that can be obtained through the distribution factor $(D_{FRP})$ given by [40].

$$f_{f,e} = E_f \varepsilon_{f,e} = E_f \varepsilon_{max} D_{FRP}$$

(12)

in which $\varepsilon_{f,e}$ is the effective strain in FRP wrap, $E_f$ is the FRP modulus of elasticity, and $D_{FRP}$ is the distribution factor obtained from Equation (10).

As shown in Figure 13a–f, the strain distributions along the shear crack are illustrated by four displacement levels corresponding to four phases: (1) the crack initiates at mid-depth of the web, (2) all fibres intercepted by shear cracks are in active phase (experiencing strain) just before initiation of delamination at the tip of the crack, (3) delamination at the cohesive layer is already formed and developed at mid-distance of shear crack paths (effective bond lengths start to decrease), and (4) one of the fibres intercepted by the shear crack is exposed to maximum strain during the loading process.

Regarding the series of small specimens, as displacements reached 2.23 mm and 2.79 mm in specimens strengthened with one and two layers, respectively, the shear crack initiated at mid-distance of the shear crack path from the end of the normalized distance. Then it propagated toward the bottom edge of the beams and the edge of the intersection between webs and flanges. When the displacements at the loading points reached 3.92 mm and 3.81 mm, the maximum strains on CFRP fabrics were 0.00637 and 0.00246 in S.S0.1L and S.S0.2L, respectively (Figure 13a,b). At this stage, shear cracks had completely formed, and all the fibres intersected by shear cracks were in the active phase (from the tips to the ends of the shear crack). The strain values on fibres intersected by the shear crack on the top part of the crack then dropped suddenly to zero due to delamination and to the short bond length compared to the bottom part of the crack. During that process, the cracks at the edge of the intersection between the flange and the web propagated horizontally.

The maximum strains experienced by fibres before entirely losing the CFRP shear contribution were 0.00866 and 0.00266 in S.S0.1L and S.S0.2L, respectively. These values are in good agreement with the corresponding experimental results—that is, 0.00714 and 0.00216, corresponding to 45% and 13% of CFRP ultimate strain, respectively. Note that the values of strains on fibres obtained from numerical analysis are larger than those obtained from experimental tests because dynamic implicit analysis was used to solve the models from which the amplified strains were recorded, whereas such an amplification did not exist in the static analysis [40]. Furthermore, strain gauges installed on EB-CFRP fabrics measure the average strains, which are lower than the maximum strain obtained from FEA [35]. For the medium beams, shear cracks appeared at mid-distance of the shear crack path when displacements at the loading points reached 3.36 mm and 3.15 mm in beams strengthened with one and two layers of CFRP fabric, respectively. When the displacements reached 4.73 mm and 5.44 mm in M.S0.1L and M.S0.2L, respectively (Figure 13c,d), the main diagonal shear cracks in both specimens became complete, and at this stage, all

fibres crossed by shear cracks (from the tips to the ends of the cracks) experienced stress and strain.

The maximum strains experienced by the fibres just before delamination were 0.0032 and 0.0033 in M.S0.1L and M.S0.2L, respectively, as presented in Figure 13c,d, representing 20% and 21% of the ultimate strain of the fibres. After these maximum strains were reached, an inactive zone where more fibres lost their contribution to shear strengthening (zero strain) developed at the support. The maximum strains obtained from numerical analysis were in good agreement with experimental results (i.e., 0.00248 and 0.0027 in M.S0.1L and M.S0.2L, respectively). The same scenario was observed for the large specimens, from the initiation of shear cracks to the complete failure of EB-CFRP. Therefore, all fibres were in active modes as complete shear cracks formed, and at this stage, displacements reached 7.26 mm and 8.27 mm in L.S0.1L and L.S0.2L, respectively (Figure 13e,f)), just before delamination.

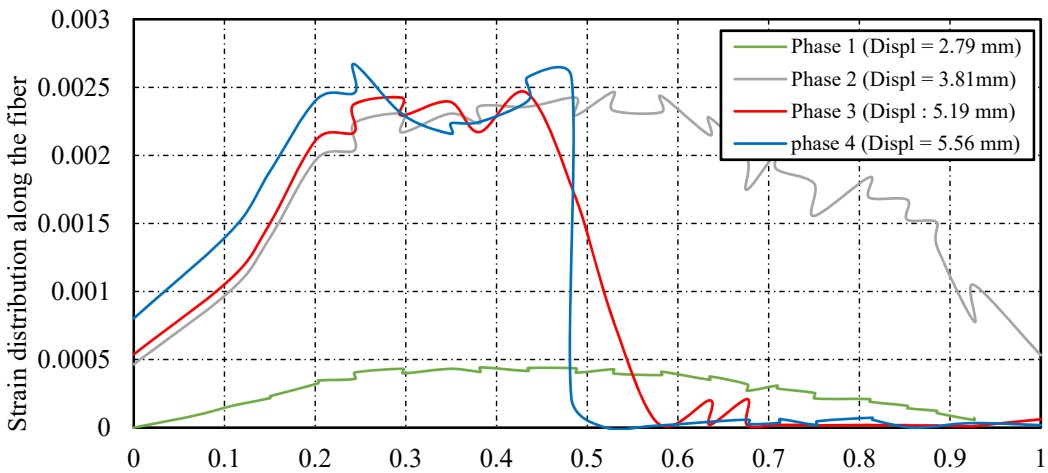

Normalized distance along the main diagonal shear crack path - Specimen S.S0.2L

(**a**)

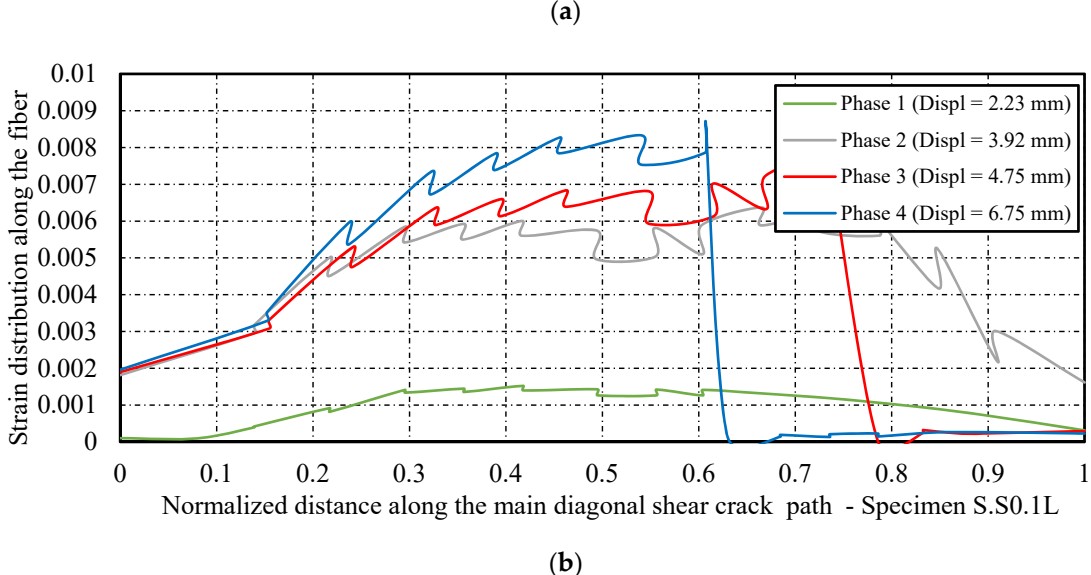

(**b**)

**Figure 13.** *Cont.*

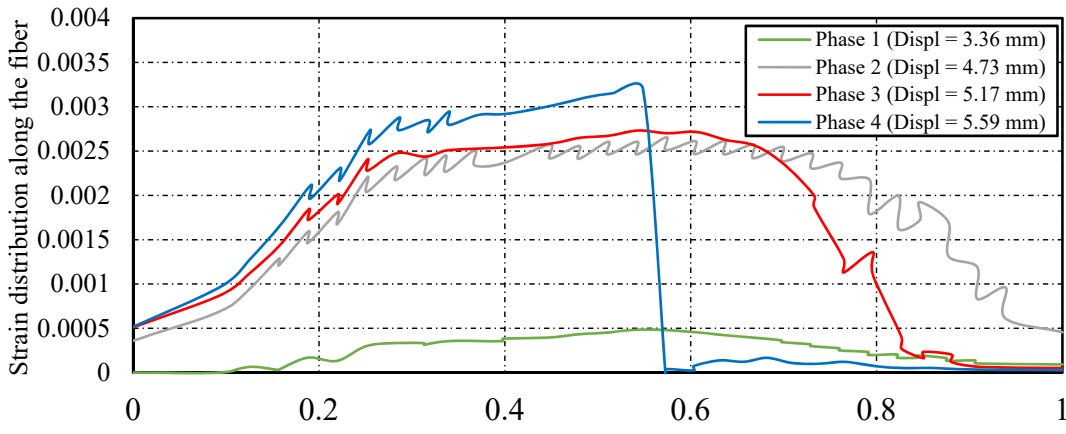

(**c**)

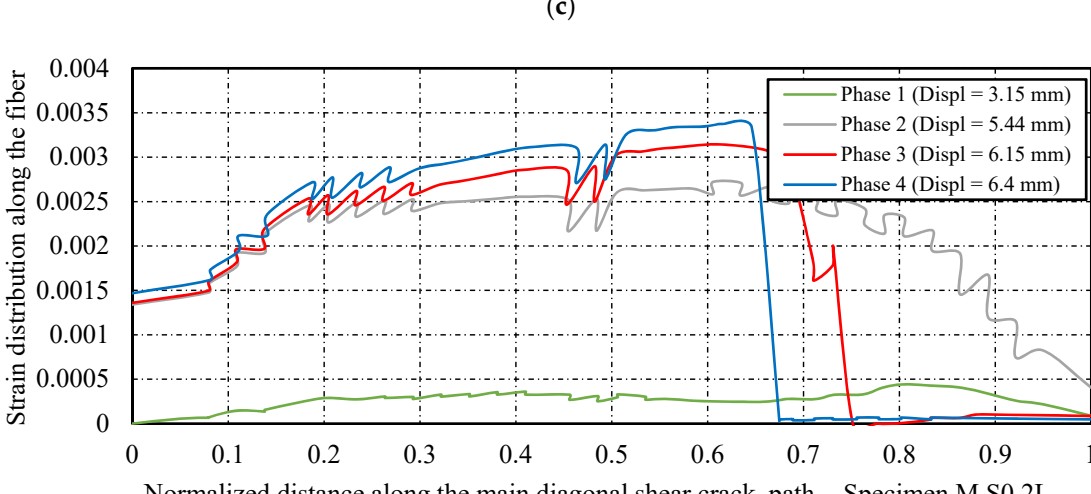

(**d**)

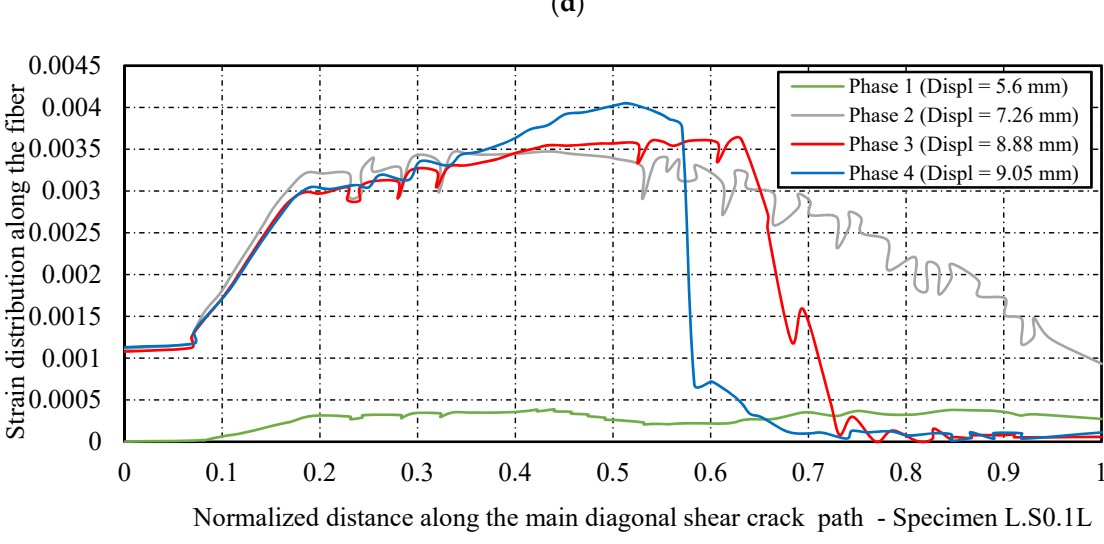

(**e**)

**Figure 13.** *Cont.*

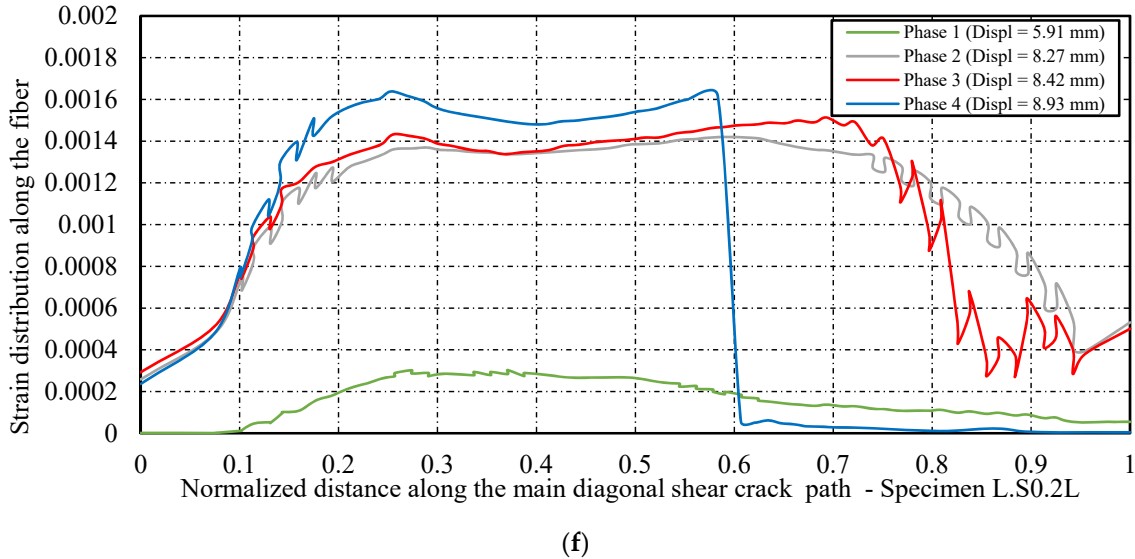

(**f**)

**Figure 13.** Distributions of strains on fibres crossed by normalized distance along the main diagonal shear path: (**a**) specimen S.S0.1L; (**b**) specimen S.S0.2L; (**c**) specimen M.S0.1L; (**d**) specimen M.S0.2L; (**e**) specimen L.S0.1L; (**f**) specimen L.S0.2L. Phase 1: initiation of main diagonal shear crack. Phase 2: all the fibres intersected by shear crack in an active phase. Phase 3: development of the loss of the shear contribution of the fibres at tips of the shear crack. Phase 4: the maximum strain recorded on fibres before the complete loss of the shear contribution of the fibres at the top part of the shear crack.

The maximum strains experienced by the fibres were 0.00415 and 0.00161 in L.S0.1L and L.S0.2L, respectively, which were very close to experimental values (0.00369 and 0.0016, respectively). Therefore, the maximum strains reached on EB-CFRP in large specimens (L.S0.1L, L.S0.2L) decreased in comparison to small beams by 53% and 40% in beams strengthened with one and two layers, respectively, resulting in a size effect on both concrete and CFRP shear contributions. Likewise, in the control beams (Figure 10a–c), the pattern of shear cracks at the final states in strengthened specimens as obtained from numerical analysis was in good agreement with experimental tests, confirming that the assumptions applied for simulation were accurate (Figure 9a–e).

*5.3. Strain Distributions along the CFRP Fabric and Interfacial Shear Stress at the Cohesive Layer*

By evaluating the strain distribution on fibres along with the normalized distance of the crack path, it is possible to locate the maximum vertical crack width. Those fibres that experience more strain before losing their shear contributions are located at the maximum crack width. Moreover, the vertical width of the crack can be calculated by summation of the interfacial slip along the two sides of the crack and the deformation of the CFRP fabric in the debonding area [35]. Therefore, after the maximum crack width was located and calculated, the strain distribution and the interfacial shear stress along the fibre intersected by the shear crack at its maximum width were evaluated to further investigate the size effect on the shear contribution of EB-CFRP.

The FEA strain profile along the fibre and the shear stress profile along the interface layer for L.S0.1L and L.S0.2L are presented in Figure 14a,b and Figure 15a,b, respectively. The results are presented in terms of strain development along the fibre direction and the interfacial shear stress along the interface layer in which debonding can be observed. Each graph shows the strain distributions and the interfacial shear stress at six displacement stages, in which the first three steps are related to initiation and development of the shear crack just before delamination, and the next three steps represent the initiation of delamination to complete loss of strain in the fibre. This yields six curves corresponding to six levels of displacement. As shown in Figure 14a,b and Figure 15a,b, there is a similarity

between the strain distribution along the main fibre direction and the strain response obtained from the pullout tests.

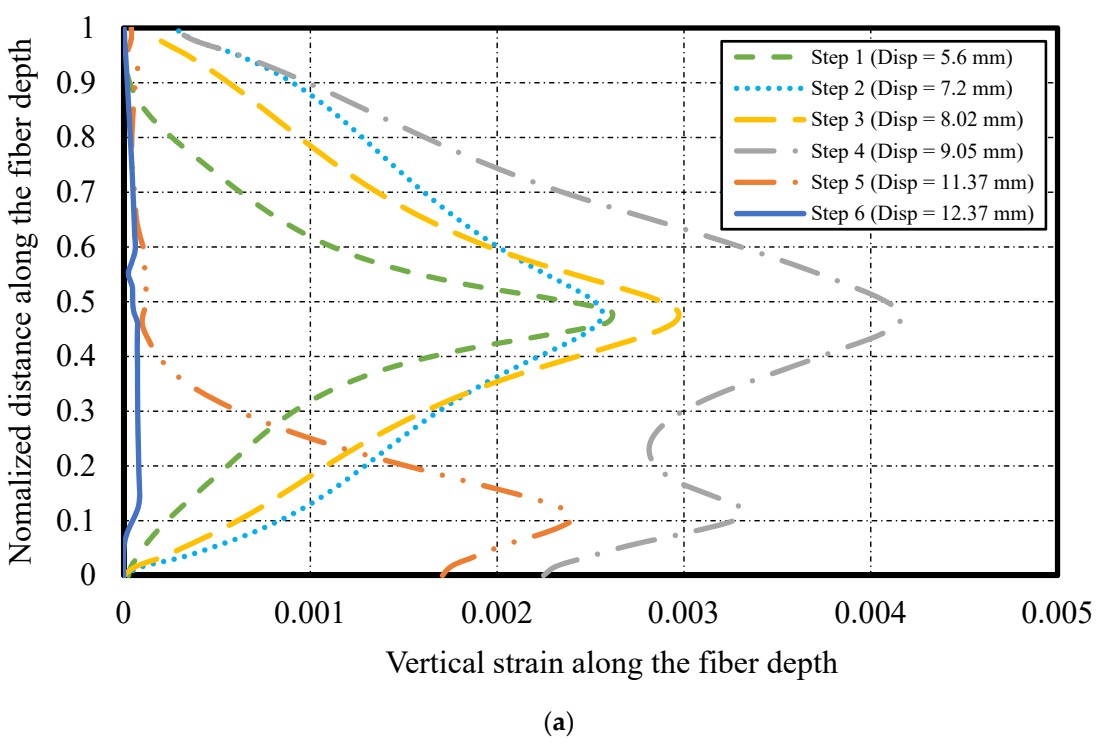

(**a**)

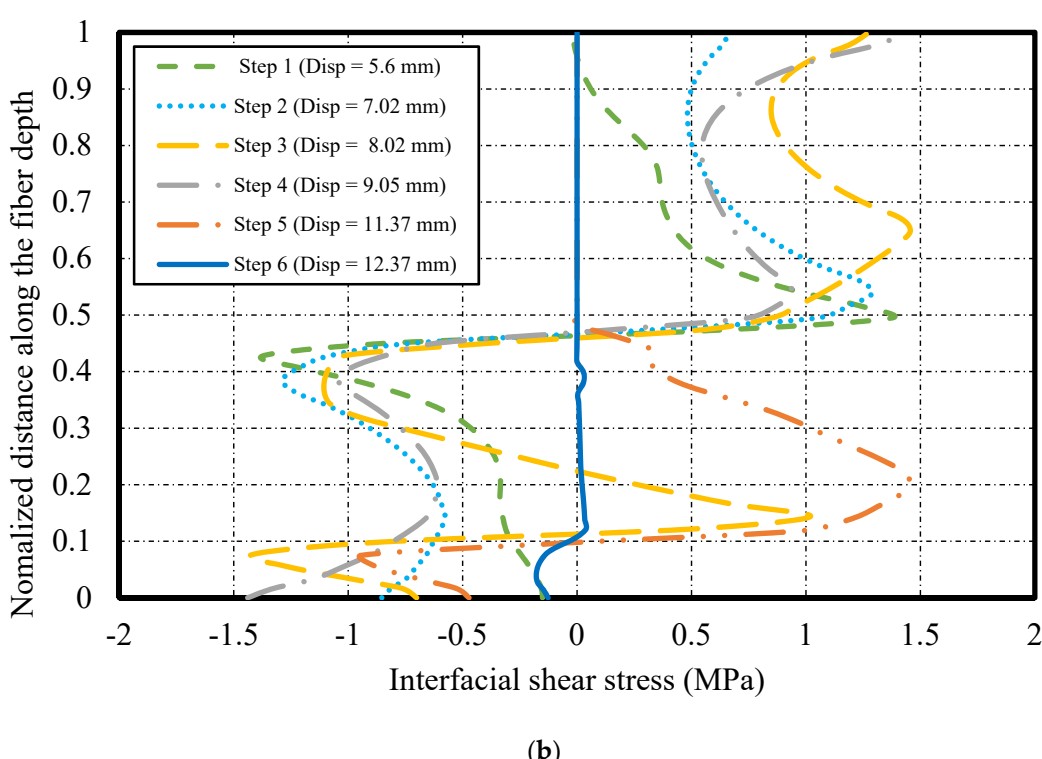

(**b**)

**Figure 14.** Strain profile and interfacial shear stress along the fibre and interface layer intercepted by maximum crack width (specimen L.S0.1L).

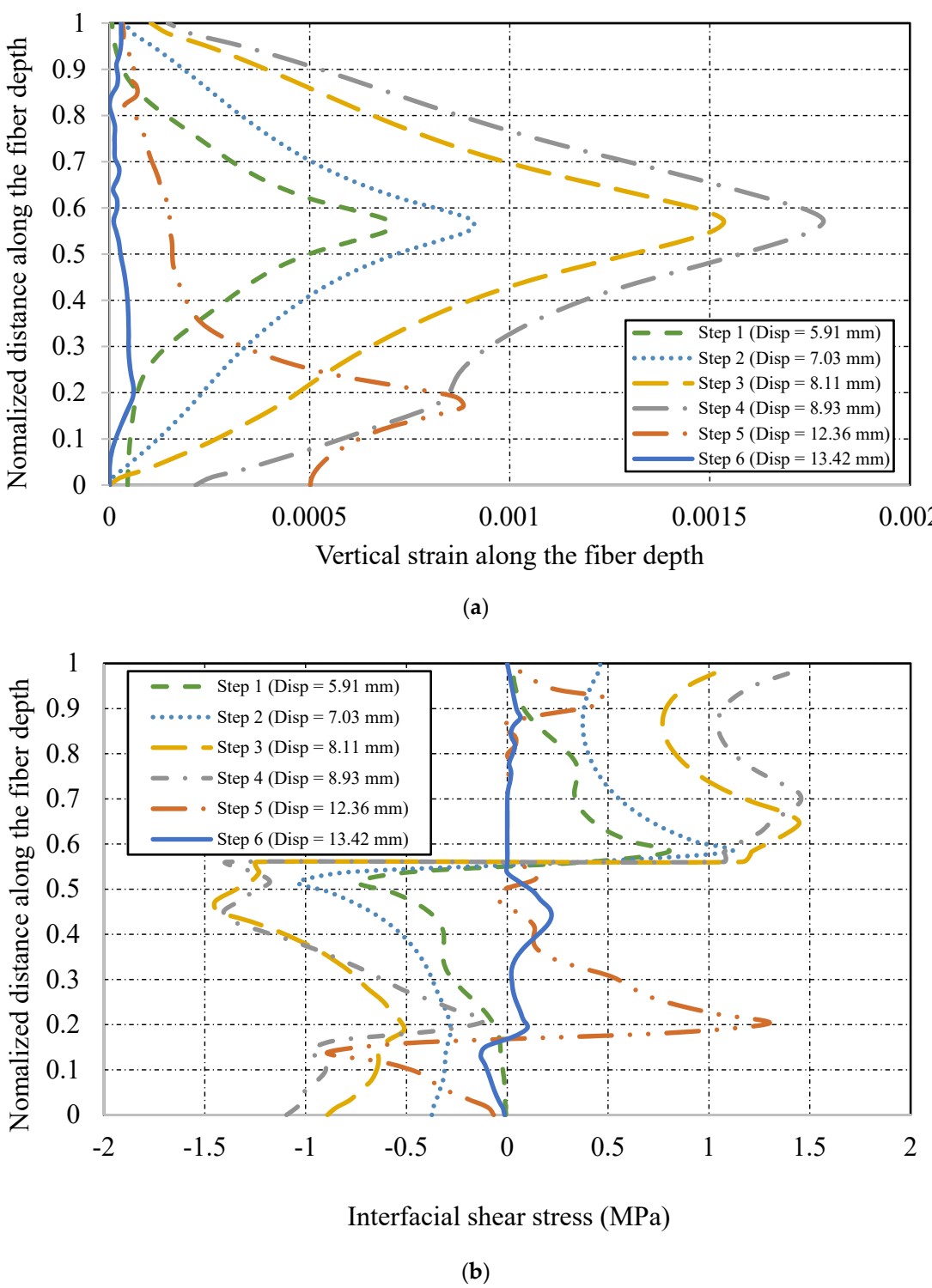

(**a**)

(**b**)

**Figure 15.** Strain profile and interfacial shear stress along the fibre and interface layer intercepted by maximum crack width (specimen L.S0.2L). Steps 1,2,3: the initiation and development of the shear crack just before the delamination procedures. Step 4,5,6: the procedure from the initiation of the delamination to the complete loss of strain in the fibre.

As soon as the shear crack appeared in L.S0.1L at 5.6 mm displacement, the fibre started to contribute to shear resistance. By increasing the load and the corresponding displacement, the amount of strain increased to a maximum of 0.00415. The maximum strain occurred at mid-depth of the specimen and then decreased gradually toward the top

edge of the web/flange intersection. The amount of strain experienced by the lower part increased more than that on the top edge of the fibre because there was more effective bond length in the bottom crack part of the U-wrap configuration, in which the fibres below the shear crack were fixed.

Note that at the peak of the strain profile and when strain was constant, interfacial shear stress was zero, indicating the delaminated zone. At the third stage, during development of the delaminated area, this zone propagated in the top part of the crack, as evidenced by zero interfacial shear stress and zero strain in the strain profile (Figures 14 and 15). As the displacement reached 12.37 mm, complete delamination occurred in specimen L.S0.L1, and likewise in specimen L.S0.2L. In addition, the amount of strain on the fibres along the main diagonal shear crack decreased in all specimens strengthened with two CFRP layers compared to specimens strengthened with one CFRP layer. Unlike the medium specimens, where increasing rigidity did not significantly change the maximum strains on the fibres, the maximum strains in all the small and large specimens (strengthened with one and two CFRP layers) decreased with increasing size, indicating the existence of an additional size effect on CFRP shear contribution. However, there is a need for more investigations regarding the relation between the size effect and the increase in CFRP rigidity. To compare maximum strain results on the contribution of CFRP fabrics to the ultimate specimen shear strength, the following dimensionless value $V_T/(b_w \times d \times f_c')$ was introduced [34]. This formula defines a dimensionless unit of the ultimate shear capacity of the beam versus the maximum strain on fibres. Therefore, it can evaluate the impact of the size effect on the ultimate specimen shear capacity (Figure 16a,b).

The maximum vertical strains were measured at the widest parts of the shear cracks. As shown in Figure 16a,b, both small specimens (S.S0.1L, S.S0.2L) showed more shear contribution of EB-CFRP than the medium and large beams. This confirms the existence of a size effect because it was expected that by increasing beam size and consequently FRP bond length, more FRP shear contribution should be obtained. In addition, despite their longer effective bond lengths, large specimens experienced less vertical strain on the fibre than small specimens, which confirms the results of previous investigations [34,39] that as beam size increases, the shear strength contribution of CFRP decreases.

In conclusion, absorption of vertical strains through the fibres is greater in smaller than in larger specimens despite the fact that both beam sizes have the same ratio of CFRP fabric.

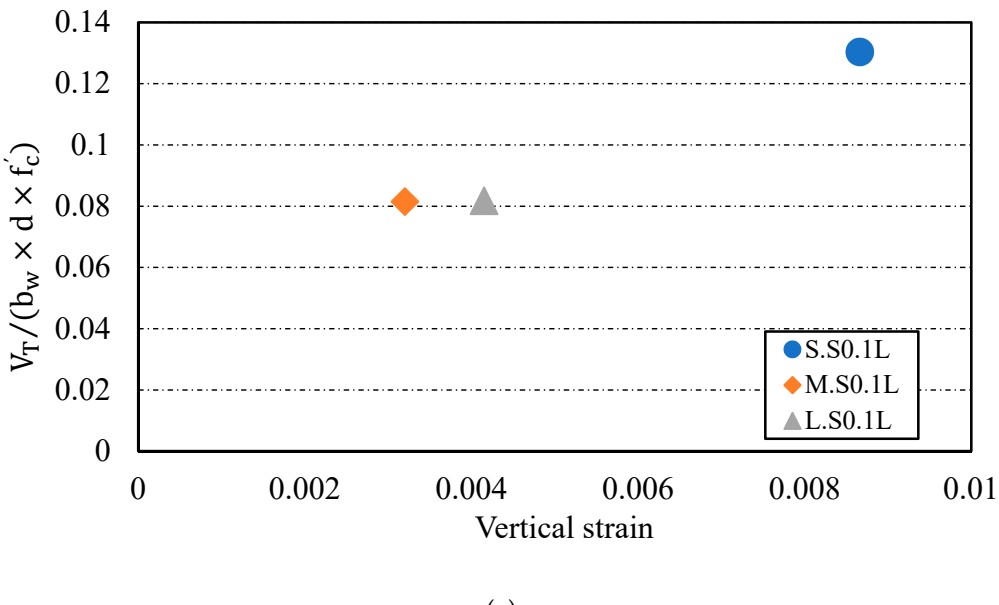

(**a**)

**Figure 16.** *Cont.*

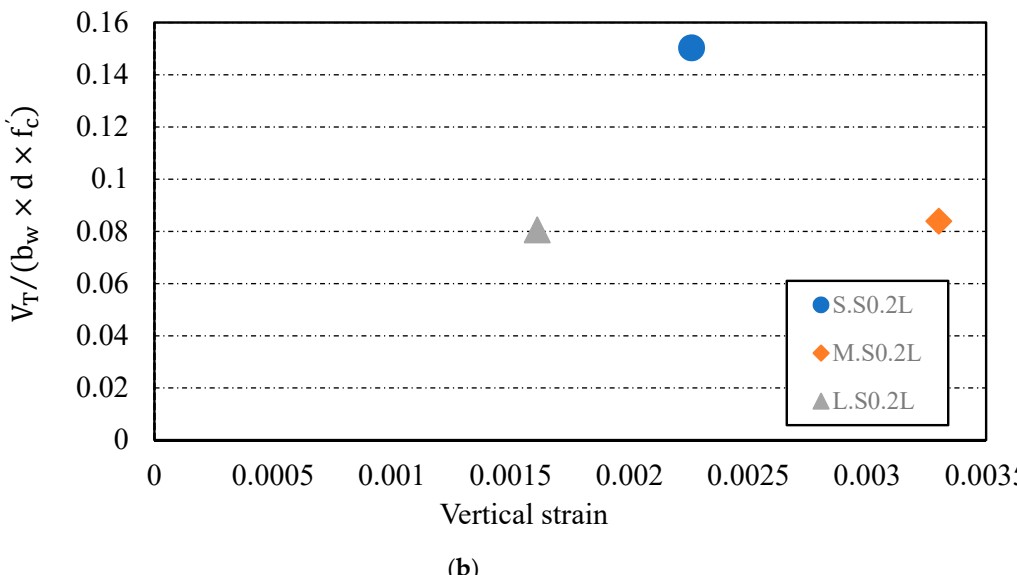

(**b**)

**Figure 16.** Correlation between maximum dimensionless shear capacity of specimens versus maximum strain along the fibre: (**a**) specimens strengthened by one layer of CFRP fabric; (**b**) specimens strengthened by two layers of CFRP fabric.

## 6. Conclusions

This research study has reported on advanced FE modelling of RC beams strengthened in shear with EB-FRP fabrics, with emphasis on the size effect on the shear contribution of RC beams shear-strengthened with EB-CFRP. The results obtained from FEA were in good agreement with the experimental results. Nine RC-T beams (three control beams and six strengthened beams in shear with EB-FRP) were simulated in ABAQUS. The results obtained from the numerical model were related to shear crack patterns, load-deflection curve, shear stress contributed by concrete and CFRP fabric, distributions of strains on fibres crossed by normalized distance along the main diagonal shear path during the loading process, and correlation between maximum dimensionless shear capacity of specimens versus maximum strain along the fibres, demonstrating that the proposed FEA is capable of capturing the response of the RC-T beams with high accuracy if the assumptions are defined properly. Furthermore, compared to experimental tests, FEA provided more precise observations and parameters during loading. The following conclusions can be drawn from this study:

- By increasing the rigidity of EB-CFRP in small specimens, the shear contribution of EB-CFRP showed a considerable increment, but in large specimens, EB-CFRP experienced a reduction in its absorption of shear stress due to the size effect.
- Delamination on the top parts of the diagonal shear cracks was the dominant failure mode compared to debonding, especially in medium and large specimens strengthened with EB-CFRP fabric.
- A reduction factor to account for size effect is of paramount importance. The reduction factor could be incorporated either into the effective strain or into the distribution factor, which are included in the model to express the shear contribution of EB-FRP.
- The delamination failure initiated suddenly around the tips of the shear crack, where the bond length was minimal.
- Considering strain profiles and interfacial shear stress along with the fibres and the interface layers, when the strain profile reached its peak value or became constant, the interfacial shear stress became zero.

**Author Contributions:** Conceptualization, A.A., Z.E.A.B., O.C. and G.E.-S.; methodology, A.A., Z.E.A.B., O.C. and G.E.-S.; software, A.A., Z.E.A.B., O.C. and G.E.-S.; validation, A.A., Z.E.A.B., O.C. and G.E.-S.; formal analysis, A.A.; investigation, A.A.; resources, O.C.; writing—original draft preparation, A.A.; writing—review and editing, O.C. and G.E.-S.; supervision, O.C. and G.E.-S.; project administration, O.C.; funding acquisition, O.C. All authors have read and agreed to the published version of the manuscript.

**Funding:** O.C. is funded by the National Science and Engineering Council (NERC) of Canada and by the Fonds de Recheche du Quebec-Nature & Technologie (FRQ-NT).

**Institutional Review Board Statement:** Not applicable.

**Informed Consent Statement:** Not applicable.

**Data Availability Statement:** The data supporting the finding of this study are available within the article.

**Acknowledgments:** The financial support of the Natural Sciences and Engineering Research Council of Canada (NSERC) and the Fonds de recherche du Québec–Nature et technologie (FRQNT) through operating grants is gratefully acknowledged.

**Conflicts of Interest:** The authors declare no conflict of interest.

## Appendix A

**Table A1.** Experimental studies assessing the shear contribution of EB-FRP by ACI 440.2R 2017.

| Specimen | Section | $d_{fv}$ | $h_f$ | $h_w$ | $b_w$ | $d$ | $nt_f$ | $n$ | $t_f$ | $a/d$ | $f'_c$ | $\rho_{FRP}(\%)$ | $\rho_w(\%)$ | $\rho_s(\%)$ | Configuration | $\beta$ | Fibre | $E_s$ | $E_{FRP}$ | $f_t$ (Mpa) | $\varepsilon_{fu}$ | $V_f$(ACI)(N) | $V_f$(exp)(N) |
|---|---|---|---|---|---|---|---|---|---|---|---|---|---|---|---|---|---|---|---|---|---|---|---|
| | | | | | | | | | | | | [52] | | | | | | | | | | | |
| T4S4-G90 | T- | 180 | 150 | 250 | 140 | 330 | 1.8 | 1 | 1.8 | 3 | 29 | 2.6 | 2.3 | 0.1 | Ct-U | 90 | G | 200 | 17.7 | 250 | 0.014124 | 45,878 | 49,000 |
| T6S4-G90 | T- | 380 | 150 | 450 | 140 | 530 | 1.8 | 1 | 1.8 | 2.8 | 44 | 2.6 | 2.7 | 0.1 | Ct-U | 90 | G | 200 | 17.7 | 250 | 0.014124 | 96,854 | 110,000 |
| | | | | | | | | | | | | [53] | | | | | | | | | | | |
| SB-U1 | Rect | 155 | | 180 | 75 | 155 | 0.11 | 1 | 0.11 | 2.5 | 27.4 | 0.1 | 5.4 | 0.28 | St-U | 90 | C | 200 | 235 | 4200 | 0.02 | 10,684 | 24,000 |
| MB-U1 | Rect | 305 | | 360 | 150 | 305 | 0.22 | 2 | 0.11 | 2.5 | 27.4 | 0.1 | 4.4 | 0.28 | St-U | 90 | C | 200 | 235 | 4200 | 0.02 | 42,049 | 5000 |
| LB-U2 | Rect | 660 | | 720 | 300 | 660 | 0.44 | 4 | 0.11 | 2.5 | 27.4 | 0.1 | 4.1 | 0.28 | St-U | 90 | C | 200 | 235 | 4200 | 0.02 | 181,984 | 22,000 |
| | | | | | | | | | | | | [54] | | | | | | | | | | | |
| S-Str | Rect | 305 | | 368 | 203 | 305 | 0.4191 | 1 | 0.4191 | 3 | 25.2 | 0.05 | 0.16 | 0 | St-U | 90 | C | 200 | 228 | 3450 | 0,015132 | 77,378 | 47,000 |
| M-Str | Rect | 457 | | 546 | 305 | 457 | 0,4191 | 1 | 0.4191 | 3 | 32 | 0.05 | 0.16 | 0 | St-U | 90 | C | 200 | 228 | 3450 | 0,015132 | 174,101 | 87,000 |
| L-Str | Rect | 610 | | 698.5 | 406 | 610 | 0.4191 | 1 | 0.4191 | 3 | 32 | 0.05 | 0.18 | 0 | St-U | 90 | C | 200 | 228 | 3450 | 0,015132 | 308,423 | 127,000 |
| | | | | | | | | | | | | [55] | | | | | | | | | | | |
| ED2 S0-1L | T- | 100 | 55 | 165 | 95 | 175 | 0.066 | 1 | 0.066 | 3 | 25 | 0.14 | 3.76 | 0 | Ct-U | 90 | C | 200 | 231 | 3650 | 0.015801 | 12,196 | 23,000 |
| ED1 S0-1L | T- | 228 | 102 | 304 | 152 | 350 | 0.066 | 1 | 0.066 | 3 | 25 | 0.14 | 3.61 | 0 | Ct-U | 90 | C | 200 | 231 | 3650 | 0.015801 | 27,808 | 39,000 |
| ED2 S1-1L | T- | 100 | 55 | 165 | 95 | 175 | 0.066 | 1 | 0.066 | 3 | 25 | 0.14 | 3.76 | 0.38 | Ct-U | 90 | C | 200 | 231 | 3650 | 0.015801 | 12,196 | 3000 |
| ED2 S0-2L | T- | 100 | 55 | 165 | 95 | 175 | 0.132 | 2 | 0.066 | 3 | 25 | 0.28 | 3.76 | 0 | Ct-U | 90 | C | 200 | 231 | 3650 | 0.015801 | 24,393 | 32,000 |
| ED1 S0-2L | T- | 228 | 102 | 304 | 152 | 350 | 0.132 | 2 | 0.066 | 3 | 25 | 0.28 | 3.61 | 0 | Ct-U | 90 | C | 200 | 231 | 3650 | 0.015801 | 55,617 | 40,000 |
| ED2 S1-2L | T- | 100 | 55 | 165 | 95 | 175 | 0.132 | 2 | 0.066 | 3 | 25 | 0.28 | 3.76 | 0.38 | Ct-U | 90 | C | 200 | 231 | 3650 | 0.015801 | 24,393 | 12,000 |
| ED1 S1-2L | T- | 228 | 102 | 304 | 152 | 350 | 0.132 | 2 | 0.066 | 3 | 25 | 0.28 | 3.61 | 0.38 | Ct-U | 90 | C | 200 | 231 | 3650 | 0.015801 | 55,617 | 4000 |
| | | | | | | | | | | | | [56] | | | | | | | | | | | |
| G1-GFRP-1B | Rect | 175 | | 200 | 100 | 175 | 1.3 | 1 | 1.3 | 1.7 | 25 | 2.6 | 1.8 | 0.19 | Ct-U | 90 | G | 200 | 26.1 | 575 | 0.022031 | 47,502 | 18,000 |
| G1-GFRP-2A | Rect | 350 | | 400 | 200 | 350 | 2.6 | 1 | 2.6 | 1.7 | 25 | 2.6 | 1.8 | 0.19 | Ct-U | 90 | G | 200 | 26.1 | 575 | 0.022031 | 190,008 | 55,000 |
| G1-GFRP-3A | Rect | 525 | | 600 | 300 | 525 | 3.9 | 1 | 3.9 | 1.7 | 25 | 2.6 | 1.8 | 0.19 | Ct-U | 90 | G | 200 | 26.1 | 575 | 0.022031 | 427,518 | 64,000 |
| G2-GFRP-2A | Rect | 442 | | 500 | 200 | 442 | 2.6 | 1 | 2.6 | 2 | 23.5 | 2.6 | 2.4 | 0.16 | Ct-U | 90 | G | 200 | 26.1 | 575 | 0.022031 | 239,953 | 80,000 |
| G2-GFRP-3A | Rect | 682 | | 750 | 300 | 682 | 3.9 | 1 | 3.9 | 2 | 23.5 | 2.6 | 2.4 | 0.16 | Ct-U | 90 | G | 200 | 26.1 | 575 | 0.022031 | 555,366 | 180,000 |
| G2-CFRP-1 | Rect | 196 | | 250 | 100 | 196 | 1 | 1 | 1 | 2 | 23.5 | 2 | 2.4 | 0.16 | Ct-U | 90 | C | 200 | 95.8 | 986 | 0.010292 | 150,214 | 25,000 |
| G2-CFRP-2 | Rect | 442 | | 500 | 200 | 442 | 2 | 1 | 2 | 2 | 23.5 | 2 | 2.4 | 0.16 | Ct-U | 90 | C | 200 | 95.8 | 986 | 0.010292 | 677,497 | 120,000 |
| | | | | | | | | | | | | [57] | | | | | | | | | | | |
| 4 | Rect | 89 | | 102 | 114 | 85 | 1.539 | 1 | 1.539 | 3.53 | 42.9 | 2.7 | 2.61 | 0 | Ct-U | 45 | G | 200 | 16.7 | 342.35 | 0.0205 | 25,879 | 31,500 |
| | | | | | | | | | | | | [58] | | | | | | | | | | | |
| CO2 | Rect | 260 | | 305 | 150 | 260 | 0.165 | 1 | 0.165 | 3.6 | 20.5 | 0.088 | 4 | 0 | St-U | 90 | C | 200 | 228 | 3648 | 0.016 | 31,299 | 40,000 |
| CO3 | Rect | 260 | | 305 | 150 | 260 | 0.165 | 1 | 0.165 | 3.6 | 20.5 | 0.22 | 4 | 0 | Ct-U | 90 | C | 200 | 228 | 3648 | 0.016 | 78,249 | 65,000 |
| | | | | | | | | | | | | [59] | | | | | | | | | | | |
| BT2 | T- | 260 | 100 | 305 | 150 | 360 | 0.165 | 1 | 0.165 | 3 | 35 | 0.22 | 2.28 | 0 | Ct-U | 90 | C | 200 | 228 | 3648 | 0.016 | 78,249 | 65,000 |
| BT3 | T- | 260 | 100 | 305 | 150 | 360 | 0,165 | 1 | 0.165 | 3 | 35 | 0.44 | 2.28 | 0 | Ct-U | 90 | C | 200 | 228 | 3648 | 0.016 | 78,249 | 67,500 |
| BT4 | T- | 260 | 100 | 305 | 150 | 360 | 0.165 | 1 | 0.165 | 3 | 35 | 0.088 | 2.28 | 0 | St-U | 90 | C | 200 | 228 | 3648 | 0.016 | 31,299 | 72,000 |

**Table A1.** *Cont.*

| | | | | | | | | | | | | | | | | | | | | | | | |
|---|---|---|---|---|---|---|---|---|---|---|---|---|---|---|---|---|---|---|---|---|---|---|---|
| | | | | | | | | | | | | | | | | | | | | | | [60] | |
| SO3-2 | Rect | 260 | | 305 | 150 | 260 | 0.165 | 1 | 0.165 | 3 | 27.5 | 0.088 | 4.2 | 0 | St-U | 90 | C | 200 | 228 | 3648 | 0.016 | 31,299 | 54,000 |
| SO3-3 | Rect | 260 | | 305 | 150 | 260 | 0.165 | 1 | 0.165 | 3 | 27.5 | 0.132 | 4.2 | 0 | St-U | 90 | C | 200 | 228 | 3648 | 0.016 | 46,949 | 56,500 |
| SO3-4 | Rect | 260 | | 305 | 150 | 260 | 0.165 | 1 | 0.165 | 3 | 27.5 | 0.22 | 4.2 | 0 | Ct-U | 90 | C | 200 | 228 | 3648 | 0.016 | 78,249 | 67,500 |
| SO4-2 | Rect | 260 | | 305 | 150 | 260 | 0.165 | 1 | 0.165 | 4 | 27.5 | 0.088 | 4.2 | 0 | St-U | 90 | C | 200 | 228 | 3648 | 0,016 | 31,299 | 62,500 |
| | | | | | | | | | | | | | | | | | | | | | | [52] | |
| T6NS-C45 | T- | 390 | 150 | 450 | 140 | 540 | 0.7 | 1 | 0.7 | 2.9 | 44.1 | 2.6 | 2.81 | 0 | St-U | 45 | C | 200 | 230 | 3450 | 0.015 | 355,193 | 103,500 |
| T6S4-C90 | T- | 390 | 150 | 450 | 140 | 540 | 0.7 | 1 | 0.7 | 2.9 | 44.1 | 0.08 | 2.81 | 0.1 | St-U | 90 | C | 200 | 230 | 3450 | 0.015 | 251,160 | 85,250 |
| T6S4-G90 | T- | 390 | 150 | 450 | 140 | 540 | 1.8 | 1 | 1.8 | 2.9 | 44.1 | 2.6 | 2.81 | 0.1 | Ct-U | 90 | G | 200 | 17.7 | 265.5 | 0.015 | 99,403 | 109,900 |
| T6S4-Tri | T- | 390 | 150 | 450 | 140 | 540 | 2.1 | 1 | 2.1 | 2.9 | 44.1 | 3 | 2.81 | 0.1 | Ct-U | 45 | G | 200 | 8.1 | 121.5 | 0.015 | 75,054 | 129,200 |
| T4NS-G90 | T- | 190 | 150 | 450 | 140 | 340 | 1.8 | 1 | 1.8 | 3.2 | 29 | 2.6 | 4.46 | 0 | Ct-U | 90 | G | 200 | 17.7 | 265.5 | 0.015 | 48,427 | 43,600 |
| T4S4-G90 | T- | 190 | 150 | 450 | 140 | 340 | 1.8 | 1 | 1.8 | 3.2 | 29 | 2.6 | 4.46 | 0.1 | Ct-U | 90 | G | 200 | 17.7 | 265.5 | 0.015 | 48,427 | 48,650 |
| T4S2-C45 | T- | 190 | 150 | 450 | 140 | 340 | 0.7 | 1 | 0.7 | 3.2 | 29 | 0.08 | 4.46 | 0.2 | St-U | 45 | C | 200 | 230 | 3450 | 0.015 | 173,043 | 17,800 |
| T4S2-G90 | T- | 190 | 150 | 450 | 140 | 340 | 1.8 | 1 | 1.8 | 3.2 | 29 | 2.6 | 4.46 | 0.2 | Ct-U | 90 | G | 200 | 17.7 | 265.5 | 0.015 | 48,427 | 24,350 |
| T4S2-Tri | T- | 190 | 150 | 450 | 140 | 340 | 2.1 | 1 | 2.1 | 3.2 | 29 | 3 | 4.46 | 0.2 | Ct-U | 45 | G | 200 | 8.1 | 121.5 | 0.015 | 36,564 | 41,400 |
| | | | | | | | | | | | | | | | | | | | | | | [61] | |
| G5.5-1L | T- | 254.1 | 88.9 | 355.6 | 92 | 343 | 0.10902 | 1 | 0.10902 | 2 | 37.9 | 0.237 | 3.6 | 1.1 | Ct-U | 90 | C | 203 | 231 | 3696 | 0.016 | 51,193 | 31,150 |
| G5.5-2L | T- | 254.1 | 88.9 | 355.6 | 92 | 343 | 0.21804 | 2 | 0.10902 | 2 | 37.9 | 0.475 | 3.6 | 1.1 | Ct-U | 90 | C | 203 | 231 | 3696 | 0.016 | 102,386 | 53,400 |
| G8-1L | T- | 254.1 | 88.9 | 355.6 | 92 | 343 | 0.10902 | 1 | 0.10902 | 2 | 37.9 | 0.237 | 3.6 | 0.76 | Ct-U | 90 | C | 203 | 231 | 3696 | 0.016 | 51,193 | 31,150 |
| G8-2L | T- | 254.1 | 88.9 | 355.6 | 92 | 343 | 0.21804 | 2 | 0.10902 | 2 | 37.9 | 0.475 | 3.6 | 0.76 | Ct-U | 90 | C | 203 | 231 | 3696 | 0.016 | 102,386 | 62,300 |
| G8-3L | T- | 254.1 | 88.9 | 355.6 | 92 | 343 | 0.32706 | 3 | 0.10902 | 2 | 37.9 | 0.712 | 3.6 | 0.76 | Ct-U | 90 | C | 203 | 231 | 3696 | 0.016 | 153,579 | 84,550 |
| G16-1L | T- | 254.1 | 88.9 | 355.6 | 92 | 343 | 0.10902 | 1 | 0.10902 | 2 | 37.9 | 0.237 | 3.6 | 0.38 | Ct-U | 90 | C | 203 | 231 | 3696 | 0.016 | 51,193 | 40,050 |
| G16-2L | T- | 254.1 | 88.9 | 355.6 | 92 | 343 | 0.21804 | 2 | 0.10902 | 2 | 37.9 | 0.475 | 3.6 | 0.38 | Ct-U | 90 | C | 203 | 231 | 3696 | 0.016 | 102,386 | 84,550 |
| G24-1L | T- | 254.1 | 88.9 | 355.6 | 92 | 343 | 0.10902 | 1 | 0.10902 | 2 | 37.9 | 0.237 | 3.6 | 0.25 | Ct-U | 90 | C | 203 | 231 | 3696 | 0.016 | 51,193 | 53,400 |
| G24-2L | T- | 254.1 | 88.9 | 355.6 | 92 | 343 | 0.21804 | 2 | 0.10902 | 2 | 37.9 | 0.475 | 3.6 | 0.25 | Ct-U | 90 | C | 203 | 231 | 3696 | 0.016 | 102,386 | 48,950 |
| G24-3L | T- | 254.1 | 88.9 | 355.6 | 92 | 343 | 0.32706 | 3 | 0.10902 | 2 | 37.9 | 0.712 | 3.6 | 0.25 | Ct-U | 90 | C | 203 | 231 | 3696 | 0.016 | 153,579 | 53,400 |

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
