# Peer review of "FE Modelling and Simulation of the Size Effect of RC T-Beams Strengthened in Shear with Externally Bonded FRP Fabrics"

_jcs, doi:10.3390/jcs6040116_

Round 1

Reviewer 1 Report

This paper presents an interesting research topic on current engineering issues. The work shows a good level of content, however, in order to be published, it requires supplementing the content with necessary remarks, as well as making some corrections:
1. in the introduction citations should be in the form [1], [2] etc. and not as (name, year).
2. in the introduction, please cite the work on composite materials made of CFRP (10.1016/j.compositesb.2020.107931, 10.1016/j.compstruct.2021.113598, 10.1016/j.compositesb.2013.10.080)
3) At the end of the introduction, please clearly emphasize the novelty of this paper in relation to other thematically similar research papers.
4) Figure 1 in the introduction should be much clearer.
5) Formulas in chapter 2 are presented a bit chaotically, please unify the notation, center them, etc.
6. figures 4-7 and 11 should be much more sharpened to make them more readable.
7. the conclusions should refer to both quantitative and qualitative evaluation of the research results.

Author Response

Response to Reviewer 1 are provided in attached file.

Reviewer 2 Report

The paper investigated the size effect on the shear behaviour of RC beams strengthen with EB-CFRP. The content of the paper is quite comprehensive and many works done which can be significant enough for publication. Before being accepted the paper should address some major concerns:

  1. Lines 127-130, the authors stated that the static solvers cannot capture the nonlinearity of materials during imposed targeted displacement. The statement is not appropriate since many complicated RC structure models are still using the implicit static and obtaining good results [1, 2]. Please revise the statement to make it more objective.
  2. Section 2.1.1 has some confusion, it mentioned the crack band model and smeared cracking approach, and figure 2 illustrated the cohesive law of stress transferring through the cracks. Those aspects are used for modelling the softening behaviour of concrete and are totally different from the rotating and fixed smeared crack model in Lines 146 and 148. Please remove those lines since the authors adopted Concrete Damage Plasticity (CDP).
  3. In the damage rules in Section 2.2.3, , what is the meaning of h ? element size, or constant length?
  4. According to with the assumption of h = element size, therefore the softening curve is mesh-size dependent. Limiters should be considered, however, the papers did not recommend the range of limiters. In the simulation, the size of elements was chosen between 5mm and 10mm, what is the reason?
  5. Some references adopted CDP models to simulate the FRP-RC structures, the review believes that those refs are beneficial to the paper.

[1] Tran D, Pham T, Hao H, Chen W. Numerical Study on Bending Response of Precast Segmental Concrete Beams Externally Prestressed with FRP Tendons. Eng Struct. 2021;(Accepted).

[2] Tran DT, Pham TM, Hao H, Chen W. Numerical investigation of flexural behaviours of precast segmental concrete beams internally post-tensioned with unbonded FRP tendons under monotonic loading. Eng Struct. 2021;249:113341.

Author Response

See attached file for response to reviewer 2.

Reviewer 3 Report

Overall, I think it's a good paper.
The actual verification of the destruction model has been sufficiently conducted and it is judged to be a model that properly reflects the actual phenomenon.
There are a few comments about future research, so please check them out.

1. In the process of reinforcing through CFRP, it seems that additional analysis of the bonding force between CFRP and steel is necessary. I think the degree of influence will vary depending on the type of structural connection.

2. It would be nice to add step-by-step images where energy concentration occurs in the destruction mode evaluation through FEA. I think it will be a better material if there are pictures that can confirm the actual destruction growth mode.

3. It would be nice if the actual model parameters could be expanded to derive data that could simulate a slightly more diverse environment. 

Author Response

See attached file for response to reviewer 3.
